# APOBEC affects tumor evolution and age at onset of lung cancer in smokers

Tongwu Zhang [1,15], Jian Sang[1,15], Phuc H. Hoang [1], Wei Zhao [1], Jennifer Rosenbaum[2], Kofi Ennu Johnson[3], Leszek J. Klimczak [4], John McElderry[1], Alyssa Klein [1], Christopher Wirth [5], Erik N. Bergstrom[6], Marcos Díaz-Gay [6], Raviteja Vangara [6], Frank Colon-Matos[1], Amy Hutchinson[1,7], Scott M. Lawrence [1,7], Nathan Cole[1,7], Bin Zhu [1], Teresa M. Przytycka [8], Jianxin Shi [1], Neil E. Caporaso[1], Robert Homer [9], Angela C. Pesatori [10,11], Dario Consonni [11], Marcin Imielinski[3], Stephen J. Chanock [1], David C. Wedge [5], Dmitry A. Gordenin [12], Ludmil B. Alexandrov [6], Reuben S. Harris [13,14] & Maria Teresa Landi [1] ✉

Most solid tumors harbor somatic mutations attributed to off-target activities of APOBEC3A (A3A) and/or APOBEC3B (A3B). However, how APOBEC3A/B enzymes affect tumor evolution in the presence of exogenous mutagenic processes is largely unknown. Here, multi-omics profiling of 309 lung cancers from smokers identifies two subtypes defined by low (*LAS*) and high (*HAS*) APOBEC mutagenesis. LAS are enriched for A3B-like mutagenesis and *KRAS* mutations; HAS for A3A-like mutagenesis and *TP53* mutations. Compared to LAS, HAS have older age at onset and high proportions of newly generated progenitor-like cells likely due to the combined tobacco smoking- and APOBEC3A-associated DNA damage and apoptosis. Consistently, HAS exhibit high expression of pulmonary healing signaling pathway, stemness markers, distal cell-of-origin, more neoantigens, slower clonal expansion, but no smoking-associated genomic/epigenomic changes. With validation in 184 lung tumor samples, these findings show how heterogeneity in mutational burden across co-occurring mutational processes and cell types contributes to tumor development.

Somatic mutations in cancer are caused by both exogenous and endogenous mutational processes[1]. Mutational signature analysis has been widely used to reveal the mutagenic processes in cancer genomic studies[2–4]. Varying levels of APOBEC mutational signatures (a combination of COSMIC mutational signatures SBS2 and SBS13 or an enrichment with a known APOBEC mutational motif TCW; W=adenine/thymine) have been reported in ~70% of human cancer types[3,5–7]. These signatures have been associated with mutational tumor heterogeneity[8] and an improved response to immunotherapy[9]. APOBEC mutagenesis is the off-target effect of the activity of the APOBEC family of enzymes that function as cytosine deaminases[10]. This off-target effect has been

predominantly attributed to APOBEC3A (A3A) and APOBEC3B (A3B)[11], which are involved in virus and retroelement restriction[12,13], response to DNA damage, and other cellular functions[14,15]. Both enzymes have access to the nucleus and are associated with elevated mRNA levels in tumors as compared with matched normal tissue[5,16]. While A3A and A3B imprint similar patterns of mutations on somatic genomes, the two enzymes exhibit enrichments for different motifs. Specifically, an enrichment of YTCA has been associated with APOBEC3A, whereas an enrichment of RTCA has been attributed to APOBEC3B (Y = pyrimidine; R = purine)[17]. The occurrence of most APOBEC-associated mutations appears to be episodic[18–20], and this process is likely driven

by the APOBEC3A deaminase[7]. APOBEC-associated deamination of cytosines can occur exclusively in single-stranded (ss)DNA[14]. APOBEC mutagenesis in tumors is enriched in the lagging strand template and in early-replicating, gene-dense, and active chromatin genome regions[21–23]. APOBEC3-catalyzed mutagenesis can also lead to chromosomal instability[24,25]. Despite considerable information accumulated about off-target mutagenesis by different APOBECs in tumors, their roles in tumor evolution in the context of other mutagenic processes remain largely unknown.

Lung cancer is the second most common cancer type worldwide and the leading cause of cancer death[26]. Tobacco smoking is the foremost risk factor for lung cancer and has strong mutagenic activity leading to high tumor mutational burden (TMB), COSMIC mutational signatures SBS4, DBS2, ID3, and, more recently SBS92[3,27,28], and increased mutation frequency in cancer driver genes[29–31]. Lung cancer also harbors APOBEC signatures in about 35 to 55% of tumors[3,6], possibly in response to DNA damage and local inflammation (likely induced by tobacco smoking)[15,32]. However, how the co-occurrence of APOBEC and tobacco smoking mutagenesis impacts lung cancer evolution has not been investigated.

Here, we present a multi-omics study including newly sequenced deep whole-genome (73x coverage), transcriptome, and epigenomic profiles of 309 paired tumor-normal lung tissue samples from smokers with detailed clinical and smoking history information. With validation in another independent dataset, we found that the co-occurrence of APOBEC and tobacco smoking mutational processes affects the development and age at onset of lung cancer. A similar effect is observed in other tumor types associated with exposure to exogenous mutagens, like melanoma.

## Results

We analyzed a multi-omics dataset encompassing a total of 345 smokers with histologically confirmed lung cancer (Supplementary Data 1). While the majority of tumors show dominant tobacco smoking-associated signatures SBS4, DBS2, ID3, and a few SBS92 as expected (Supplementary Data 2), 36 tumors (10.4%) had no mutations assigned to SBS4, even in the subset of 20 (9.2%) samples from the Environment And Genetics in Lung cancer Etiology (EAGLE) study[33] with very high-confidence and detailed smoking information (Supplementary Data 3). Although some of these tumors had DBS2 and ID3 signatures, we excluded the 36 samples lacking signatures SBS4 and SBS92 from the subsequent analyses to ensure analyzing only samples from smokers. Thus, the reported analysis includes 309 samples with SBS4 signature, constituted of 286 lung adenocarcinomas (LUAD, 83%), 36 squamous cell carcinomas (LUSC, 11%), and 20 other or mixed subtypes (6%; Supplementary Fig. 1). We conducted the analyses across all samples and separately in LUAD only. In addition, we included 184 LUAD tumors from the TCGA study for validation purposes.

### APOBEC mutational signatures define two lung cancer subtypes in smokers

Analysis of mutational signatures, through SigProfilerExtractor and nonnegative matrix factorization[28], revealed two distinct groups identified by the presence (43.7%) or absence (56.3%) of APOBEC mutational signatures SBS2 and SBS13 (Fig. 1a and Supplementary Fig. 2). The absence of APOBEC signatures can be due to technical reasons, including the commonly used 5% of all mutations' threshold for attributing somatic mutations to a signature within an individual sample[28,34–36]. To confirm this hypothesis, we utilized P-MACD[5,17], a specialized orthogonal computational approach that focuses purely on detecting APOBEC trinucleotide motifs while providing a minimum estimation of APOBEC mutation load and a sample-specific $P$-value (Supplementary Data 4).

Among the "absent" APOBEC signature group, P-MACD identified as APOBEC-positive 158 samples with significant enrichment of

APOBEC mutations and 16 samples lacking an enrichment of the APOBEC trinucleotide mutational motifs (Supplementary Fig. 3a–c). In addition, P-MACD confirmed all 135 samples with APOBEC signatures identified by the SigProfilerExtractor algorithm (Pearson correlation, $R = 1$; $P = 0$). This suggests that APOBEC mutagenesis operated in all groups, however, at different levels. As a further confirmation, excluding the 16 samples with no APOBEC mutations identified by P-MACD did not substantially change the results (data not shown). Moreover, kataegis (clusters of localized hypermutation in cancer genomes) or C- or G-coordinated clusters were present in many samples (Supplementary Fig. 3d and Supplementary Data 4), including in samples with no detected APOBEC mutations by both methods. Thus, we named the tumors with APOBEC mutations detected by P-MACD or not detected by any method as "Low APOBEC Subtype" (LAS) and those identified by both methods as "High APOBEC Subtype" (HAS). Compared to LAS, HAS tumors showed a significantly higher frequency of kataegis ($P = 7.5e-03$; Supplementary Fig. 3d), overall number of mutations within kataegis ($P = 0.014$; Supplementary Fig. 3e), as well as a higher proportion of APOBEC mutations contributing to kataegis (61.4% versus 50.8%; Supplementary Fig. 3f, g). Importantly, HAS tumors were dominated (70.4%) by the A3A-like mutator phenotype[17], whereas LAS tumors were dominated (69%) by the A3B-like phenotype ($P = 1.1e-48$; Fig. 1b).

No significant differences were observed between LAS and HAS in terms of stage, histology, sex, tobacco smoking phenotype variables, germline APOBEC3B deletion, tumor purity, TMB, percentage of genome altered (PGA) by copy numbers, number of structural variations (SV), or SBS4 mutation burden (Fig. 1c and Supplementary Fig. 4). The frequency of nonsynonymous mutations in TP53 ($P = 0.03$; Fig. 1d and Supplementary Fig. 5a, b) and a few other driver genes ($P < 0.05$; e.g., PIK3CA and ERBB2; Fig. 1d and Supplementary Fig. 6) is higher in HAS compared to LAS ($P = 7.28e-03$, $3.21e-09$, and $4.28e-05$ for TP53, PIK3CA, and ERBB2, respectively), in line with the association between APOBEC activity and driver gene mutations reported in other studies[37–39]. We found very similar results when we restricted the cancer driver gene nonsynonymous mutations to those defined as drivers based on the Cancer Genome Interpreter platform (**Methods**; Supplementary Fig. 7). As expected for APOBEC-related mutations, we found a significant enrichment of hotspot C > X mutations (X = any base) at TpC sites in these genes (e.g., TP53: 22.8% in HAS, 14.2% in LAS, $P = 1.37e-07$; OR = 1.77). Unlike HAS tumors, LAS were enriched with KRAS driver mutations ($P = 1.1e-03$; Fig. 1d and Supplementary Fig. 5d, e). We confirmed the same findings with TP53 ($P = 0.041$) and KRAS ($P = 0.00042$) enrichments in HAS and LAS tumors, respectively, when we restricted the analyses to LUAD only (Supplementary Fig. 5c, f). We further validated these findings in the TCGA LUAD whole-exome sequencing dataset and observed consistent results with strong enrichment of TP53 in HAS ($P = 4.67e-04$), and KRAS in LAS ($P = 8.04e-03$; Supplementary Fig. 8) tumors. In addition, HAS tumors exhibited a significantly higher number of retrotransposon insertions ($P = 0.026$; Fig. 1e), as expected for a tumor subtype with higher genomic instability[40]. In contrast, we found no mutation difference in SMUG1 and REV1, which have been shown to be directly linked to the generation of APOBEC3-mediated mutational signatures[41].

### Association between APOBEC3B and UNG expression

Among all AID/APOBEC genes, RNA-Seq analyses showed that only APOBEC3A and APOBEC3B had significantly higher expression in HAS compared to LAS ($P = 1.95e-03$ and $2.27e-04$, respectively; Fig. 2a and Supplementary Data 5). As expected, APOBEC3A and APOBEC3B did not differ in normal tissue. Notably, only APOBEC3A expression was significantly associated with APOBEC mutation burden in HAS tumors, whereas only APOBEC3B expression was significantly associated with APOBEC mutations in LAS tumors ($R = 0.24$ and $0.46$; $P = 0.028$ and $3.6e-06$, respectively; Fig. 2b).

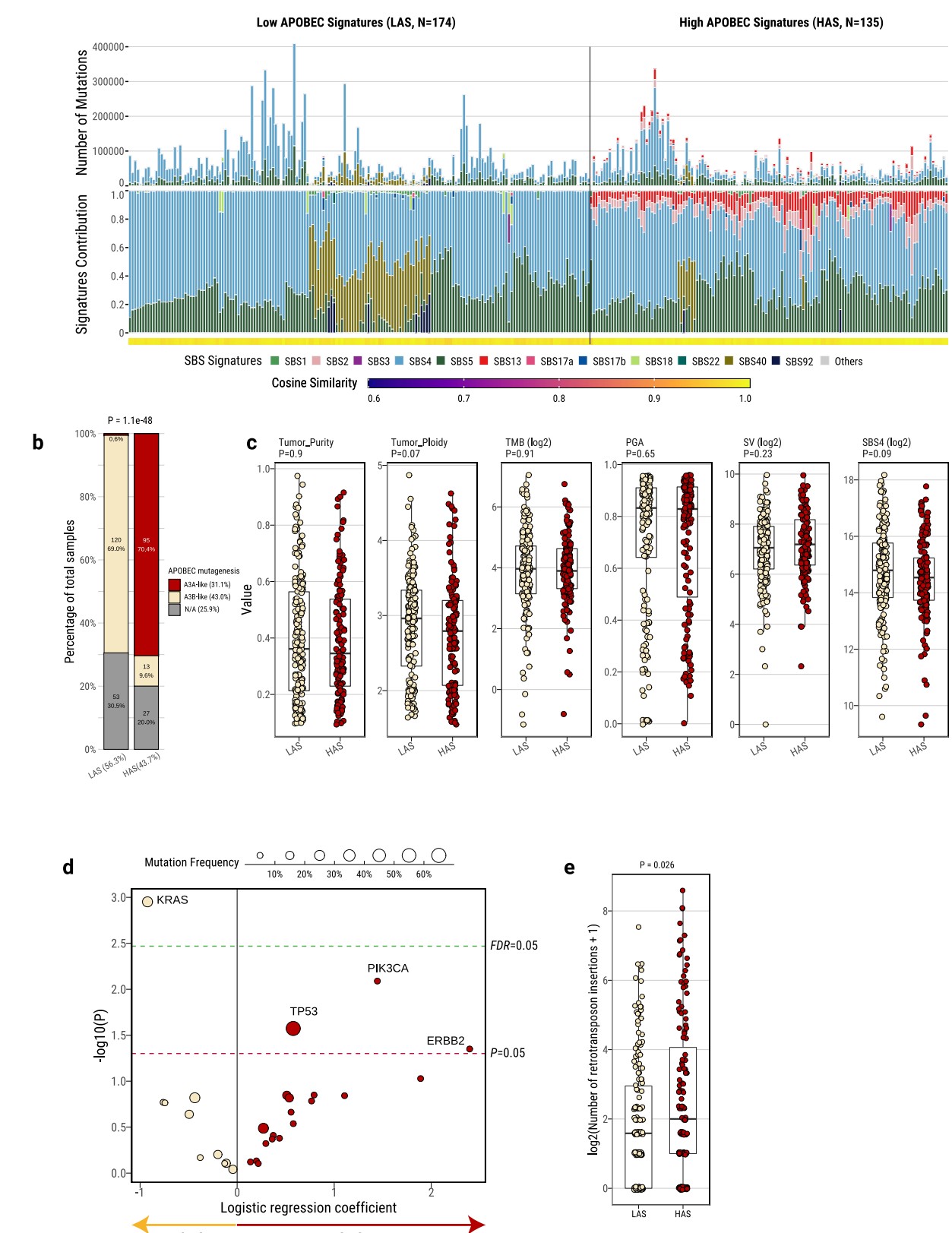

**a** Low APOBEC Signatures (LAS, N=174) / High APOBEC Signatures (HAS, N=135)

SBS Signatures: SBS1, SBS2, SBS3, SBS4, SBS5, SBS13, SBS17a, SBS17b, SBS18, SBS22, SBS40, SBS92, Others

Cosine Similarity: 0.6 0.7 0.8 0.9 1.0

**b** P = 1.1e-48

APOBEC mutagenesis: A3A-like (31.1%), A3B-like (43.0%), N/A (25.9%)

**c** Tumor_Purity P=0.9; Tumor_Ploidy P=0.07; TMB (log2) P=0.91; PGA P=0.65; SV (log2) P=0.23; SBS4 (log2) P=0.09

**d** Mutation Frequency 10% 20% 30% 40% 50% 60%

Enriched in LAS — Enriched in HAS

**e** P = 0.026

APOBEC-catalyzed C-to-U lesions can be repaired by base excision repair (BER)[42,43], primarily by uracil N-DNA glycosylase (UNG, also called UNG2 for the nuclear enzyme). Thus, we investigated the relationship between *UNG* and *APOBEC3A/B* gene expression. *UNG* gene expression was significantly upregulated in HAS compared to LAS in all samples (*P* = 0.0086; Supplementary Fig. 9a) and in samples that were copy number neutral at *UNG* genomic location (*P* = 0.02; Supplementary Fig. 9b). A significant and positive correlation of gene expression was found between *UNG* and *APOBEC3B* in both tumor subtypes (LAS: *R* = 0.35, *P* = 4.75e-04, and FDR = 5.23e-03; HAS: *R* = 0.46, *P* = 8.52e-06, and FDR = 9.37e-05), but not between *UNG* and *APOBEC3A* (LAS: *R* = −0.042, *P* = 0.68 and FDR = 1; HAS: *R* = 0.088, *P* = 0.42 and FDR = 1;

**Fig. 1 | Genomic classification and characterization of lung cancer in smokers based on mutational signatures analyses. a** Landscape of SBS mutational processes and identification of two tumor subtypes based on APOBEC mutational signatures. The landscape of mutational signatures includes a bar plot presenting the total number of mutations assigned to each signature, the proportion of signatures assigned to each sample, and the cosine similarity between the original mutation profile and the signature decomposition. **b** Proportions of A3A-like and A3B-like mutagenesis between LAS and HAS tumors. Tumors not enriched with TCA mutations or without significant differences between RTCA and YTCA mutations are classified as N/A. The *P*-values derived from the two-sided Chi-squared test are shown above the plots. **c** Comparison of genomic alterations and features between LAS (*n* = 174 tumors) and HAS (*N* = 135 tumors). The *P*-values derived from the two-sided Wilcoxon rank-sum test are shown above the plots. **d** Logistic regression analysis between tumor subtypes and nonsynonymous mutation status of driver genes, adjusting for the following covariates: age, sex, histology, TMB, and tumor purity. The significance thresholds *P* < 0.05 (red) and *FDR* < 0.05 (green) are indicated by the dashed lines. Multiple testing correction was performed using the Benjamini–Hochberg method. **e** Number of retrotransposon insertions in LAS (n = 174 tumors) and HAS (*n* = 135 tumors). The *P*-values derived from the two-sided Wilcoxon rank-sum test are shown above the plots. All box plots display the median (centerline), interquartile range (box), and whiskers extending to 1.5 × the interquartile range (IQR) by default in ggplot2. Each data point is plotted individually as a dot. Source data are provided as a Source Data file.

Fig. 2c; FDR correction based on the 32 tested genes in the BER pathway). No association was observed between *UNG* and *APOBEC3A* or *APOBEC3B* in normal tissue (Supplementary Fig. 10a). Additional genes involved in the BER pathway exhibited a similar pattern, with positive association with *APOBEC3B* but not *APOBEC3A* in tumor samples and no associations in normal samples (Supplementary Fig. 11). We observed a similar pattern with *UNG* positively associated with *APOBEC3B*, but not *APOBEC3A* in TCGA LUAD tumor samples (Supplementary Fig. 10b, c) as well as in multiple cancer types in TCGA based on RNA-Seq data (*e.g.*, prostate adenocarcinoma; head and neck squamous cell carcinoma; pancreatic ductal adenocarcinoma; lymphoid neoplasm diffuse large B-cell lymphoma; lung squamous cell carcinoma; Fig. 2d). We note that 91.8% of APOBEC mutations were clonal in HAS tumors, although APOBEC subclonal mutations were higher than the subclonal mutations due to other mutational processes (Supplementary Fig. 12a, b). Similarly, we found that, although the ratio of APOBEC mutations over all mutations was higher for subclonal mutations than clonal mutations, most APOBEC mutations are clonal across different cancer types in the PCAWG study[44] (Supplementary Fig. 12c).

## Tobacco smoking addiction is associated with genomic changes only in LAS tumors

Mutational signature analysis showed that the known tobacco smoking-associated signatures (SBS4, DBS2, and ID3) are positively associated with each other and with the APOBEC signature as well as with other mutational signatures and the overall TMB (Supplementary Fig. 13), as previously shown[27]. Among the five tobacco smoking phenotype variables available in 198 subjects from the Environment and Genetics in Lung cancer Etiology (EAGLE) study[33] (Fig. 3a), we previously found that "time to first cigarette" (TTFC) in the morning, a marker of strong nicotine addiction and high tobacco smoking exposure[45], showed the only significant association with TMB and tobacco smoking signatures[46], possibly because of a decreased DNA repair function in those who wake up early to smoke[47]. However, when we separated the tumors between LAS and HAS in this study, short TTFC was associated with an increased TMB ($P_{trend}$ = 0.0024, Fig. 3b and $P_{trend}$ = 0.0054 in LUAD only, Supplementary Fig. 14; **Methods**) and the other tobacco smoking signatures (SBS4, $P_{trend}$ = 0.005; DBS2, $P_{trend}$ = 0.0044; and ID3, $P_{trend}$ = 0.0029; Supplementary Fig. 15) only in LAS samples. Moreover, only LAS tumors showed a significant association between short TTFC and increased mutational burden across different mutation types (Supplementary Fig. 16) and in frequently mutated genes known to be associated with tobacco smoking[31] (*e.g.*, *ZFHX4*; Fig. 3c, d). Surprisingly, the frequency of genetic variants associated with nicotine addiction[48] did not vary between HAS and LAS tumors (Supplementary Fig. 17), thus this genetic variation does not appear to be the cause of the LAS/HAS difference. In addition, we observed a significant interaction between TTFC and APOBEC subtypes ($P_{interaction}$ = 0.046) when assessing whether TTFC affects the relationship between APOBEC mutagenesis and driver gene mutations in *TP53*. This finding highlights the potential interplay between APOBEC activity and tobacco smoke-induced mutagenesis, an interaction also supported by a recent study on oral tumorigenesis in animal models[49].

## Tobacco smoking-induced epigenomic changes can be reversed after quitting smoking only in LAS tumors

Smoking exposure has been shown to induce DNA hypomethylation in the *AHRR* gene[50]. *AHRR* hypomethylation has been proposed to be a marker of tobacco smoking status[51] and lung cancer mortality[52]; whereas increased methylation at this locus has been used as a marker of successful smoking cessation[53–55]. Altered methylation levels have been seen in other genes following tobacco smoking exposure, although to lesser degrees[56–58]. To understand whether hypermutation-induced cell regeneration in HAS also alters the association between smoking behaviors and DNA methylation changes, we tested associations between smoking variables and methylation levels at known smoking-associated CpG probes (reviewed recently in ref. 58) in both tumor and normal datasets of HAS and LAS (Supplementary Data 6). As expected, cg05575921 (*AHRR*) was hypomethylated in current smokers *versus* former smokers and according to years from quitting smoking (Fig. 3e and Supplementary Fig. 18). Notably, in normal lung tissue, 'time since last quitting smoking' and 'total smoking duration' were associated with altered methylation levels at 25 sites (16 genes) and 8 sites (5 genes), respectively (Supplementary Fig. 19a). As expected in normal tissue, 'time since last quitting smoking' was the only variable associated with any CpG probes, particularly in cg14120703 (*NOTCH1*), cg05575921 (*AHRR*), and cg10420527 (*LRP5*), when both 'time since quitting' and 'smoking duration' were included in the same regression model (Supplementary Fig. 19b). Importantly, in tumor samples, the association between cg05575921 (*AHRR*) and smoking status or time since last quitting smoking is observed only in LAS after adjusting for age, sex, tumor histology and tumor purity (time since last quitting smoking: *P* = 3.8e-05, Fig. 3e, f; *P* = 5.81e-05 in LUAD only). In contrast, *NOTCH1* (Supplementary Fig. 20) and other genes (Supplementary Data 7) whose methylation levels are not associated with tobacco smoking in tumors showed no difference between LAS and HAS. These findings further support the hypothesis that the dynamic cell composition in HAS tumors can indirectly disrupt the reversion of methylation levels in cg05575921 (*AHRR*) following smoking cessation. We confirmed these results in TCGA LUAD samples (Supplementary Fig. 21).

## Cell senescence followed by cell regeneration in HAS tumors

To further confirm the differences in cell composition between HAS and LAS tumors, as a consequence of the enrichment of A3A-like mutagenicity and *TP53*-induced genomic instability in HAS, we analyzed RNA-Seq data reflecting the transcriptomic landscape at the time of diagnosis. We tested which gene expression pathways were associated with TTFC in the two APOBEC-based subtypes. The most significantly upregulated pathway associated with short TTFC in HAS tumors was the pulmonary healing signaling pathway (Z-score=3.78; *P* = 3.34e-04; Supplementary Fig. 22a; **Methods**), which is related to a

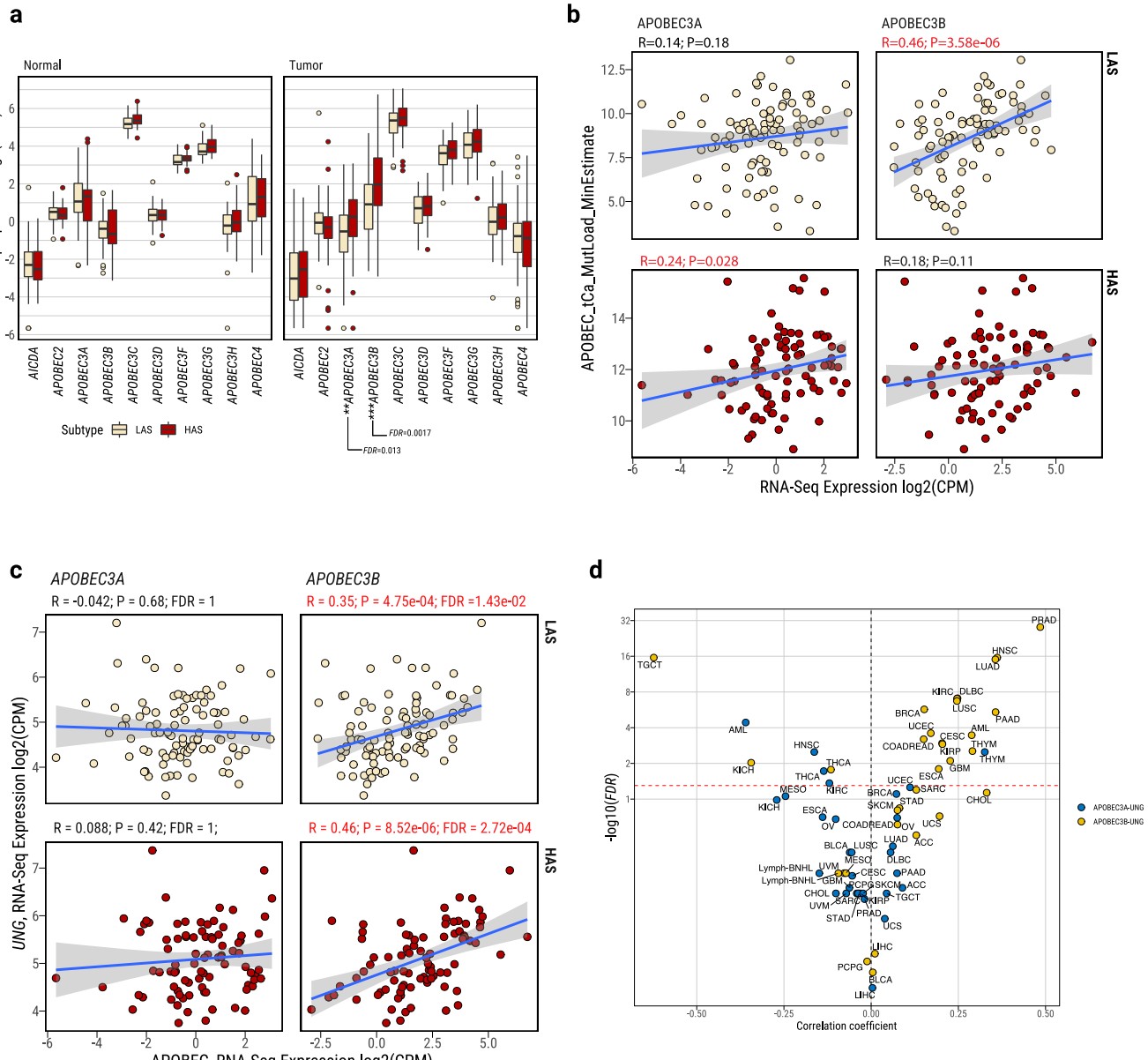

**Fig. 2 | Characterization of *APOBEC3A* and *APOBEC3B* expression in lung cancers from smokers. a** Differentially expressed APOBEC family genes between LAS and HAS in both normal and tumor samples. Sample sizes are as follows: normal tissues−LAS ($n = 45$ samples), HAS ($n = 34$ samples); tumor tissues−LAS ($n = 97$ samples), HAS ($n = 86$ samples). After multiple testing corrections based on the Benjamini−Hochberg method, only *APOBEC3A* and *APOBEC3B* show significant differential expression between LAS and HAS tumors. Of note, *APOBEC1* expression was extremely low across most tumor samples, thus it is not included in the analysis. **b** Correlation between minimal estimated APOBEC TCA mutational load from P-MACD and gene expression of *APOBEC3A* and *APOBEC3B*, stratified by LAS and HAS tumors. Pearson correlation coefficients and *P*-values are labeled above each plot and in red ink if $P < 0.05$. **c** Gene expression correlation between *UNG* and *APOBEC3A* (left) or *APOBEC3B* (right), stratified by LAS (top) and HAS (bottom)

tumors. Significant *P*-values and Pearson correlation coefficients are shown on top of each scatter plot. FDR values were calculated using the Benjamini-Hochberg method based on 32 genes in the base excision repair pathway. **d** Validation of gene expression correlation between *UNG* and *APOBEC3A* and *APOBEC3B* in all TCGA cancer types. Volcano plot shows the correlations between *UNG* and *APOBEC3A* (blue) and between *UNG* and *APOBEC3B* (yellow). The suggested significance threshold (FDR = 0.05) is indicated by a dashed red line. All box plots display the median (centerline), interquartile range (box), and whiskers extending to 1.5 × the interquartile range (IQR) by default in ggplot2. Each data point is plotted individually as a dot. Cancer type abbreviations from the TCGA study can be found here: https://gdc.cancer.gov/resources-tcga-users/tcga-code-tables/tcga-study-abbreviations. In (**b**, **c**), the shaded area represents the 95% confidence level. Source data are provided as a Source Data file.

dynamic process of regeneration of alveolar cells from stem cells. The expression of this pathway was significantly higher in HAS than LAS tumors (Z-score=1.57; $P = 1.06e\text{-}03$; Supplementary Fig. 22b). These findings suggest that HAS tumors at the time of diagnosis have a higher frequency of quiescent stem cells that have exited the quiescent state. In fact, differential expression analysis between HAS

and LAS tumors highlights elevated expression of basal cell markers (*e.g., KRT19, KRT15,* and *TP63*), indicative of lineage infidelity[59] or squamous differentiation in HAS tumors (Fig. 4a and Supplementary Fig. 22c). In contrast, LAS tumors have a higher expression of alveolar type II (AT2) cell markers (*e.g., NKX2-1, NAPSA* and *SFTPB*), which are expected in cells of the alveolar epithelia[60]. These findings were

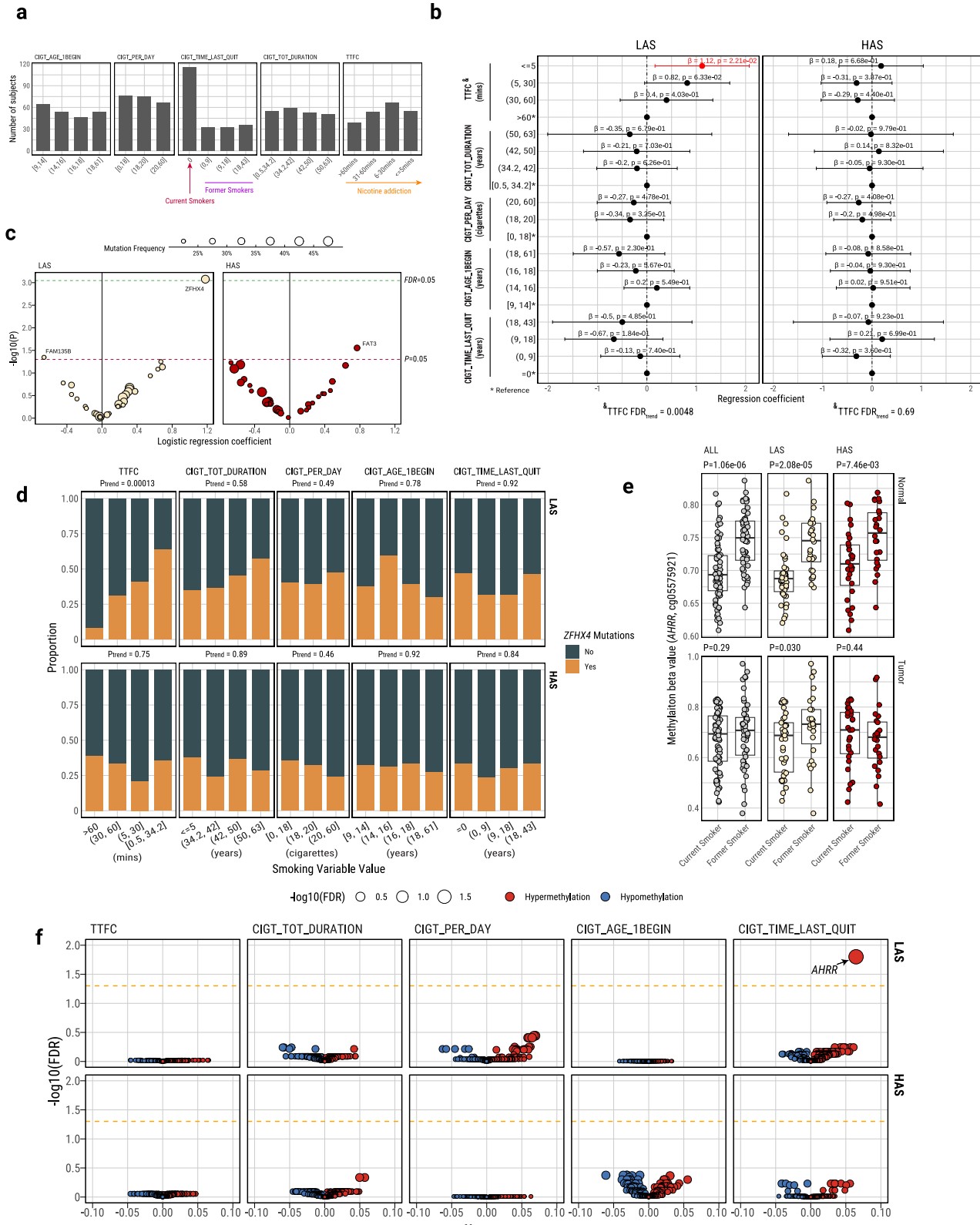

confirmed after adjustment for copy number alterations and tumor purity (Supplementary Data 8). *KRAS* mutant tumors have significantly higher expression of AT2 cell markers in both LAS and HAS tumors (Supplementary Fig. 23), in line with mouse models showing that AT2 cells are a key cell of origin for LUAD, rapidly providing cell proliferation upon induction of *KRAS* mutations[61–63]. We also inferred the cell-of-origin by correlating somatic mutational density with single cell-based lung expression profiles of benign epithelial cell types (**Methods**). Consistent with the RNA-seq results, we observed proximal lung cell types (Club/Basal/Ciliated cells) enriched in HAS and distal lung cell types (AT2) enriched in LAS (*P* = 0.045 and 8.81e-06, respectively; Supplementary Fig. 24).

**Fig. 3 | Multivariate regression analysis between five tobacco smoking variables and genomic or epigenomic features in the EAGLE samples.**
**a** Distributions of the values of each smoking variable in the 218 EAGLE samples. **b** Forest plot for the associations between TMB and smoking variables, stratified by LAS (*n* = 114 tumors) and HAS (*n* = 84 tumors). *P*-values and regression coefficients with 95% confidence intervals (CIs) are shown for each category of smoking variables. Significant associations are highlighted in red. Trend test *P*-values, adjusted for multiple testing using the Benjamini–Hochberg method (FDR$_{trend}$) from associations between TTFC and TMB are included below the forest plots. Error bars represent 95% confidence intervals of the regression coefficients. **c** Volcano plot shows the association between each TTFC category and the mutation status of commonly mutated genes (Frequency > 20%). We performed logistic regression analyses between LAS and HAS tumors. The size of each point on the volcano plot indicates the overall gene mutation frequency. The red and green dashed line indicates the association significance threshold *P* = 0.05, and FDR = 0.05, respectively. **d** Example of an association between each TTFC category and *ZFHX4* mutation frequency stratified by LAS and HAS subtypes. Trend test *P*-values (P$_{trend}$)

are labeled above each subplot. **e** Multivariate regression analysis of the DNA methylation level at CpG probe cg05575921 within the *AHRR* gene and smoking status, conducted in tumor (*n* = 116 samples) and normal (*n* = 119 samples) EAGLE tissue samples. The association analyses are performed on all tumors and separately between LAS and HAS tumor subtypes. Trend test *P*-values (P$_{trend}$) are labeled above each subplot. **f** Volcano plots of the associations between smoking variables and methylation levels of known smoking-related CpG probes (*n* = 116 tumors). Association FDR values (adjusted using the Benjamini-Hochberg method) are shown on the y-axis. The orange dashed line indicates the associations with FDR < 0.05. The CpG probes associated with tobacco smoking are derived from a study[58] comparing methylation levels between smokers and never smokers in normal lung tissue. The size and color of each point represent the FDR and association direction, respectively. All association analyses are adjusted for the following covariates: age, sex, histology, and tumor purity. All box plots display the median (centerline), interquartile range (box), and whiskers extending to 1.5 × the interquartile range (IQR) by default in ggplot2. Each data point is plotted individually as a dot. Source data are provided as a Source Data file.

## APOBEC influences age at onset and neoantigens presentation

We hypothesized that recently generated cells in HAS tumors require a long time to accumulate mutations before the tumors become clinically evident. Consistent with this hypothesis, the estimation of methylation-based mitotic rate in HAS tumors showed lower cell division than LAS tumors (*P* = 0.022; Fig. 4b). Importantly, patients with HAS tumors showed a later age at diagnosis (univariate test *P* = 0.025; *P* = 0.022 after adjusting for histology, stage, and sex; Fig. 4c) in comparison to LAS cancers. This difference between LAS and HAS tumors was strongly affected by smoking exposure levels, with LAS showing ~10 years and ~5 years younger age at onset in subjects with shorter TTFC (< = 5 min; univariate test *P* = 2.2e-04 and *P* = 0.0034 for LUAD only; multivariate test *P* = 5.91e-04 and *P* = 3.14e-03 for LUAD) or a high number of cigarettes per day (> 20; univariate test *P* = 0.038 and *P* = 0.022 for LUAD only; multivariate test *P* = 0.075 and *P* = 0.092 for LUAD), respectively (Fig. 4d). The proportion of APOBEC mutations was positively associated with age at diagnosis in HAS tumors (*R* = 0.28; *P* = 9.43e-04; Fig. 4e; *R* = 0.39; *P* = 4.21e-05 in LUAD only). We found a later age at onset of the most recent common ancestor (MRCA; Supplementary Fig. 25a, b) and no significant difference between HAS and LAS in terms of latency, measured as the difference between the age at diagnosis and the estimated patients' age at the appearance of the MRCA (Supplementary Fig. 25c, d). These findings suggest that APOBEC mutagenesis has a stronger effect on the tumor clonal expansion, precisely when tobacco smoking mutagenesis has its strongest effect[44], than on the subsequent tumor progression. In contrast, age at diagnosis was not associated with mutation status of major cancer driver genes (i.e., *KRAS* and *TP53*, Supplementary Fig. 26).

Notably, the accumulated mutations during HAS evolution exhibited a higher propensity to generate neoantigens than in LAS, adjusting for the non-synonymous mutation burden (*P* = 0.00046; Fig. 4f; *P* = 3.12e-13, Supplementary Fig. 27a in TCGA LUAD). Within HAS, APOBEC mutations generated more neoantigens than tobacco smoking (*P* = 2.5e-18). Intriguingly, HAS tumors also showed a higher *PD-L1* expression (*P* = 0.020, Supplementary Fig. 27b, and *P* = 0.0095 in TCGA HAS LUAD, Supplementary Fig. 27c), and an enriched immune and inflammatory signaling pathway in comparison with LAS tumors (Supplementary Fig. 22d). The high neoantigens and *PD-L1* expression suggest that HAS tumors not only are diagnosed later than LAS, but they may also respond better to immunotherapy[9], as also suggested by other studies[9,64,65], a treatment option that could compensate for the APOBEC3A-related resistance to targeted treatments[66]. Of note, HAS tumors also showed a not-significant improved survival than LAS tumors, even though no patients in this study received immune-related treatments (Supplementary Fig. 28).

We verified whether an association between APOBEC signature subtypes and age at diagnosis is present across different cancer types

in the PCAWG study, taking advantage of the WGS data that allows a precise classification of HAS and LAS subtypes even in tumor types with low mutational burden. Notably, the strongest difference was found in Skin-Melanoma (10 years, *P* = 0.0094), a tumor type associated with a strong exogenous mutagenic exposure (UV radiation), like lung cancer (Supplementary Fig. 29a). Since the UVR signatures SBS7a/b have an overlapping context with SBS2, we repeated the analyses separating melanoma HAS and LAS subtypes only using SBS13 and confirmed the results (Supplementary Fig. 29b). Thus, while clock-like SBS1 and SBS5 signatures have been shown to increase with age across different tissue types[3,67], our findings suggest that high APOBEC mutagenesis indirectly influences tumor age at onset in the presence of other strong exogenous mutagenic processes. Of note, we observed no genomic or age at diagnosis differences when we stratified tumors between the presence or absence of other mutational signatures (*e.g.*, SBS40; Supplementary Fig. 30).

## Validation in an Independent LUAD WGS Dataset from TCGA

We collected and analyzed the newly sequenced WGS data of LUAD from the TCGA study (Supplementary Data 9). After quality control and processing using the same bioinformatic pipelines we used for the Sherlock-*Lung* study, we identified 184 tumor samples as a validation dataset for comparison (**Methods**). Compared to our original discovery dataset, the validation dataset has a significantly lower proportion of HAS tumors (*P* = 0.004; Supplementary Fig. 31a). The lifetime exposure to tobacco smoking was also lower in the validation dataset compared to our discovery dataset (*P* = 0.041; Supplementary Fig. 31b) and has lower numbers of smoking variables (e.g., they lack data on TTFC). Despite differences in exposure and sample size, the major findings of the discovery dataset were replicated in the validation dataset. For example, HAS tumors exhibited a significantly higher number of retrotransposon insertions (Supplementary Fig. 31c), enriched *PIK3CA* and *TP53* driver mutations (Supplementary Fig. 31d), higher expression of *APOBEC3A* and *APOBEC3B* (Supplementary Fig. 31e), and higher expression of basal cell markers (*KRT15* and *TP63*) compared to LAS tumors (Supplementary Fig. 31f). Moreover, as in the original dataset, LAS tumors were enriched with *STK11* and *KRAS* driver mutations and showed higher expression of AT2 cell markers (*NKX2-1* and *SFTPB*). HAS tumors were dominated (80.4%) by the A3A-like mutator phenotype, whereas LAS tumors were dominated (51.6%) by the A3B-like phenotype (*P* = 1.62e-23; Supplementary Fig. 31g). Similarly, APOBEC3B was strongly associated with *UNG* expression in both LAS and HAS tumors (Supplementary Fig. 31h). The age at diagnosis was found to be significantly higher in HAS tumors compared to LAS tumors (median 6.5 years difference; Supplementary Fig. 31i). In addition, a positive correlation between APOBEC mutation ratio and age at diagnosis was observed in the validation set (Supplementary

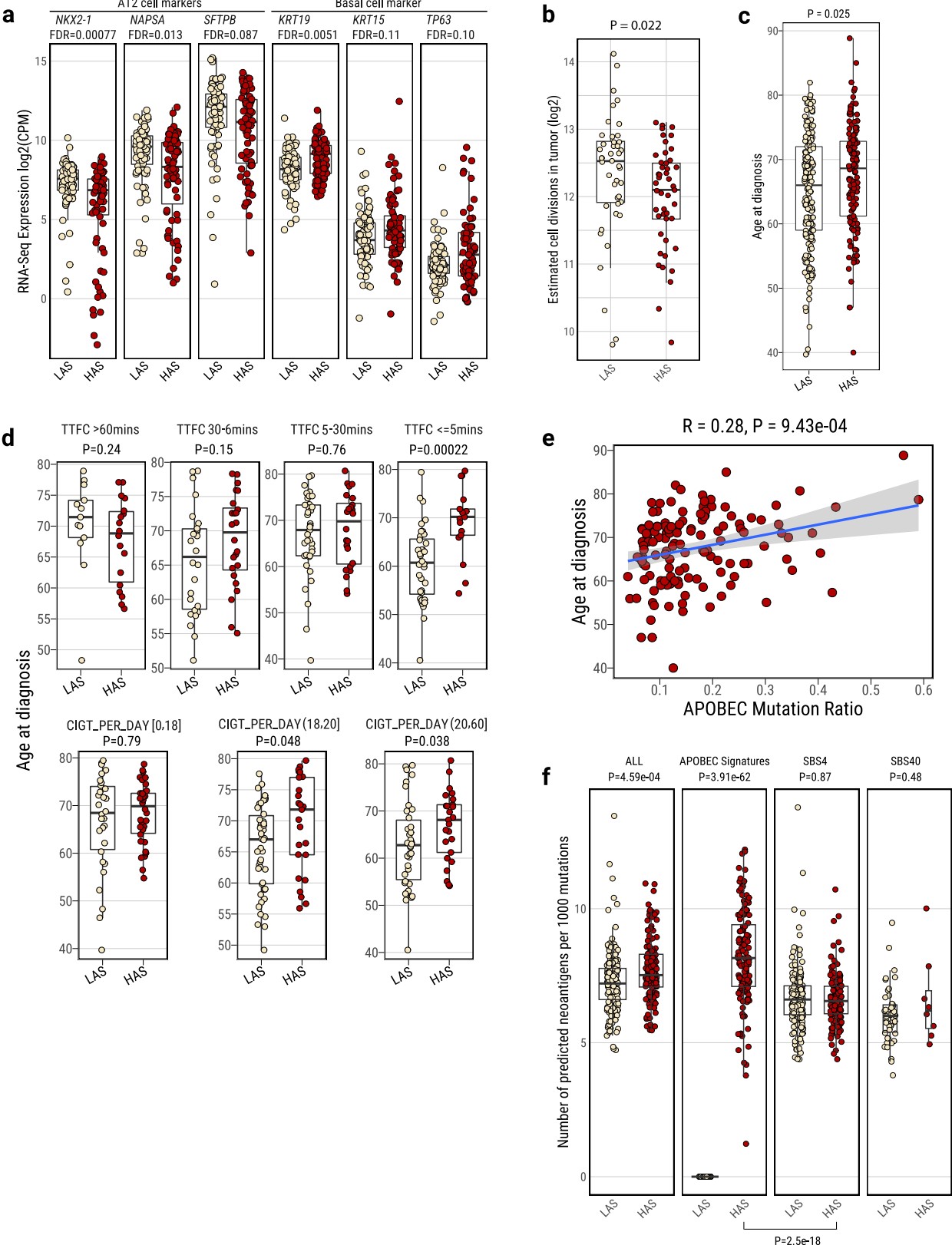

Fig. 31j). Although the validation set lacked detailed tobacco smoking information, these results strengthen the findings of genomic, transcriptomic, and cellular differences between HAS and LAS tumors and warrant further investigation into the interplay between exogenous and endogenous activities in tumor development, a phenomenon also investigated in mouse models[49].

## Discussion

Our large multi-omics study shows that the co-occurrence of APOBEC and tobacco smoking mutagenesis affects lung tumor evolution and age at onset of lung cancer from smokers (Fig. 5).

Two distinct tumor subtypes are defined by APOBEC mutagenesis. One is characterized by APOBEC signatures SBS2/13 accounting for

**Fig. 4 | Tumor cell composition and age at onset differences between LAS and HAS tumors. a** Boxplots show the differential expression of gene markers specific to lung cell types in LUAD tumors, comparing LAS ($n = 84$ tumors) and HAS ($n = 71$ tumors). **b** Cumulative number of stem cell division estimates in LAS and HAS tumors based on methylation data (LAS: $n = 43$ tumors; HAS: $n = 47$ tumors). **c**, **d** Age at diagnosis difference between LAS and HAS tumors overall (**c**), and (**d**) stratified by TTFC [Time to first cigarette in the morning (from the first question of the Fagerstrom test for nicotine dependence: 'How soon after you wake up do you smoke your first cigarette?')] or CIGT_PER_DAY (Average intensity of cigarette smoking, measured as the number of cigarettes per day). Sample sizes: overall−LAS ($n = 174$ tumors), HAS ($n = 135$ tumors); with TTFC data−LAS ($n = 112$ tumors), HAS ($n = 83$ tumors); with CIGT_PER_DAY data−LAS ($n = 114$ tumors), HAS ($n = 84$

tumors). **e** Correlation between APOBEC mutation ratio and age at diagnosis in HAS tumor. The shaded area represents the 95% confidence level. **f** Neoantigen prediction for different mutational signatures between LAS and HAS. Sample sizes: overall−LAS ($n = 174$ tumors), HAS ($n = 135$ tumors); with SBS4 signature−LAS ($n = 174$ tumors), HAS ($n = 135$ tumors); with APOBEC signature−LAS ($n = 0$ tumors), HAS ($n = 135$ tumors); with SBS40 signature−LAS ($n = 46$ tumors), HAS ($n = 8$ tumors). *P*-values from the two-sided Wilcoxon rank-sum test are labeled for each boxplot. On the bottom, *P*-value for the different contributions of SBS4 and APOBEC mutational signatures to neoantigen prediction in HAS tumors. All box plots display the median (centerline), interquartile range (box), and whiskers extending to 1.5 × the interquartile range (IQR) by default in ggplot2. Each data point is plotted individually as a dot. Source data are provided as a Source Data file.

more than 5% of mutations in each sample and strong A3A-like mutagenesis (HAS). The other has a lower level of APOBEC mutagenic activity (< 5%) and a dominant A3B-like phenotype (LAS). Both *APOBEC3A* and *APOBEC3B* expressions are highest in HAS. Intriguingly, HAS and LAS tumors are enriched with *TP53* and *KRAS* alterations, respectively.

Episodic APOBEC mutagenesis can be enhanced by ssDNA APOBEC substrate formation in response to DNA damage, such as that caused by tobacco carcinogens[68]. A3A mutagenesis, higher in HAS tumors, is known to be stronger than A3B[7,17,69]. Unlike *APOBEC3B*, *APOBEC3A* expression is not associated with mRNA expression of *UNG* and other BER genes in our study, suggesting that A3A mutations are less likely to be repaired by UNG. The frequent episodic APOBEC mutagenesis could explain the lack of association between APOBEC3A and UNG/BER expression. However, if tumors experienced frequent APOBEC bursts, we would see APOBEC mutations at different times, including during subclone expansion. The fact that many samples, particularly HAS tumors, exhibit only clonal mutations does not support this hypothesis. Moreover, it is unlikely that chronologically discrepant episodic APOBEC3A mutagenesis and UNG/BER induction show similar findings across all cancer types and multiple BER enzymes. APOBEC3A has been reported to activate DNA damage response and cause cell-cycle arrest[70]. We hypothesize that the high DNA damage due to unrepaired tobacco smoking and high A3A mutagenesis can trigger cell senescence or apoptosis in HAS tumors, in line with previous studies linking apoptosis with *TP53* alterations[71–75]. In the presence of *TP53* alterations (enriched in HAS tumors), during cellular senescence, LINE-1 retrotransposable elements (enriched in HAS) have been shown to become transcriptionally derepressed and activate a type-I interferon (IFN-I) response, which in turn contributes to sterile inflammation, a hallmark of aging[76]. Cell senescence[77], apoptosis[78], and inflammation[79] contribute to cell regeneration. Consistent with these observations, the pulmonary healing signaling pathway associated with cell regeneration showed higher expression in HAS *versus* LAS, particularly in heavy smokers. Mechanistically, single-cell transcriptomics analyses in mice models have shown that during LUAD evolution, p53 promotes AT1 differentiation through action in a transitional cell state. Notably, p53 also directs alveolar regeneration after injury by regulating AT2 cell self-renewal and promoting transitional cell differentiation into AT1 cells. In this way, wild-type p53 governs proper alveolar differentiation[80]. However, when *TP53* is mutated, these regulatory processes are disrupted, potentially preventing cells from undergoing full differentiation and thereby contributing indirectly to the more stem-like phenotype observed in the HAS (*TP53*-mutant) tumors.

Consistently, HAS tumors had higher markers of lineage infidelity and stemness than LAS. In contrast, the expression of typical LUAD AT2 cells and distal cell-of-origin were higher in LAS tumors, which were enriched with *KRAS* mutations. AT2 cells are the likely cells of origin of lung adenocarcinomas harboring *KRAS* mutations[62]. Although our study supports the AT2-like cell of origin of tumors harboring

*KRAS* mutations, further investigation on the link between AT2 cell types and KRAS mutations is warranted.

Importantly, the dynamic cellular state and composition in HAS could explain why the expected associations of tobacco smoking exposure with genomic or with epigenomic changes are not observed in this subtype. Strong DNA damage induced by APOBEC3A hypermutation and *TP53*-associated genomic instability can cause more cell senescence and apoptosis[81–84] in HAS tumors. Apoptosis and senescence can, in turn, lead to cell regeneration[85,86]. HAS tumor composition likely includes a high number of newly generated or de-differentiated cells, which do not display the expected tobacco smoking-associated patterns that are evident in the more differentiated LAS tumors. In fact, LAS and HAS tumors have similar overall mutational burdens, copy number alterations, and structural variants, although they strongly differ between A3A and A3B mutagenesis and TP53-related genomic instability. Intriguingly, studies on human bronchial epithelial cells have shown that tobacco smoking dose-dependent mutation frequency reaches a plateau at around 23 pack-years, followed by a similar mutational burden between heavy smokers and light smokers[87], possibly a consequence of frequent cell senescence/apoptosis in tumors from heavy smokers.

Previous studies suggest that APOBEC mutagenesis contributes to proteogenomic tumor evolution and heterogeneity in metastatic thoracic tumors[8]. Here, we further refine our understanding of how heterogeneity in mutational burden−driven by co-occurring mutational processes and diverse cell types−influences clonal evolution, offering important insights into lung carcinogenesis. Regenerated cells following senescence or apoptosis in HAS are likely to require a long time to accumulate mutations and to undergo subsequent clonal expansion. In addition, hypermutation and related neoantigens can trigger an immune response in the tumor microenvironment, slowing the tumor growth[88,89]. In contrast, LAS tumors, with their lower APOBEC-related mutational burdens, fewer neoantigens, and infrequent LINE 1 retrotransposons, are less likely to undergo cell senescence or apoptosis or to induce a strong immune response. Consequently, they can more steadily accumulate tobacco smoking induced mutations in the progenitor cells, increasing the probability of earlier clonal expansions. This could explain why LAS tumors have an earlier age at diagnosis than HAS tumors, particularly in the presence of strong tobacco smoking exposure, as indicated by the short time to first cigarette in the morning or a high number of cigarettes per day. We observed a similar pattern in melanoma, which is known to be associated with high UV exposure.

In conclusion, APOBEC appears to interplay with other mutagenic activities on the cancer genome[49], with important clinical implications. Although the combination of orthogonal analyses across multiple data types confirm our hypothesis, experimental validation is needed to further validate our findings. Currently, investigations into possible cancer treatment strategies inhibiting APOBEC3 mutagenesis are

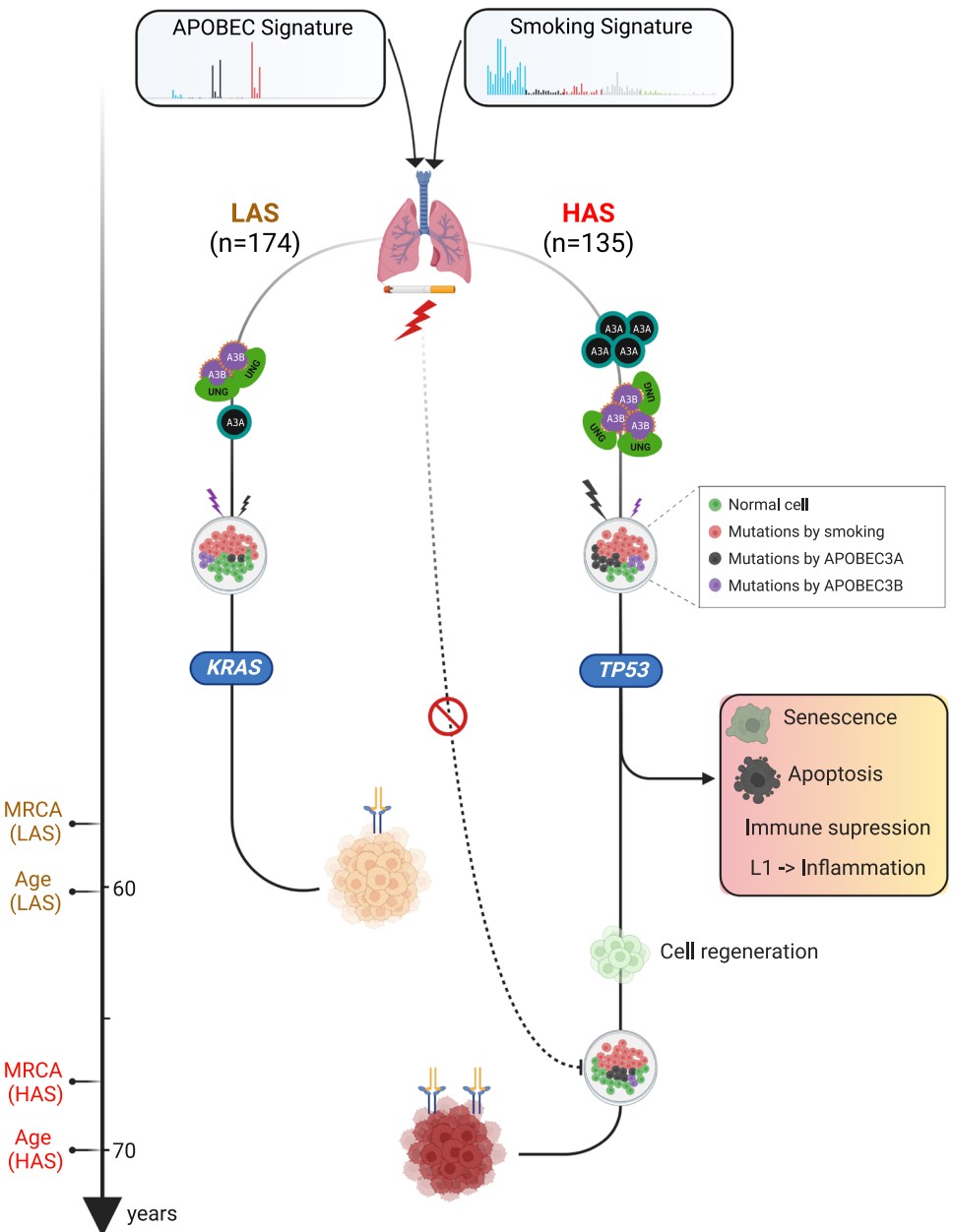

**Fig. 5 | Conceptual diagram of APOBEC shaping tumor development and influencing age at onset of lung cancers from smokers.** The schematic was generated using BioRender (https://biorender.com/).

ongoing[41]. Our findings suggest that the mutagenic contexts in which APOBEC3 operates, as well as the specific type of APOBEC mutagenesis, need to be considered in developing effective treatments.

Finally, this study emphasizes the importance of collecting detailed exposure information and WGS data in cancer studies to better understand the interplay between exogenous and endogenous activities in tumor development.

## Methods
### Data sources
We originally collected multi-omics datasets from a total of 345 smokers with pathologically confirmed and treatment-naïve lung primary cancers, including 345 tumors with WGS, 196 tumors with RNA-seq, and 206 tumors with DNA methylation. Among these tumors, 221 were from the EAGLE study[33], with 218 of these subjects having detailed clinical and smoking information. Briefly, EAGLE tumor samples were histologically confirmed as primary lung cancers with at least 50%

tumor nuclei and less than 20% necrosis. Samples were snap-frozen in liquid nitrogen within 20 min of surgical resection, and the precise site of tissue sampling was recorded. Detailed information on tumor characteristics, recurrence, treatment, and follow-up data were recovered from patients' medical records, and follow-up visits and hospital admissions were identified by linkage with the region-wide Regional Health Authority database. The study protocol was approved by the Institutional Review Board of the US National Cancer Institute and the involved institutions in Italy. Informed consent was obtained for all subjects prior to study participation. The remaining 124 smokers originated from multiple cancer genomic studies[90–96] (Supplementary Data 1). To investigate the interplay of APOBEC and tobacco smoking mutagenesis, we only included 309 tumors with detected SBS4 signatures in this analysis. The following WGS data inclusion criteria were used for sample selection before downstream somatic analyses: (1) average sequencing depth in tumor samples > 40x and normal samples > 25x; (2) sample contamination rate < 1% measured by

both Conpair (v.0.2)[97] and Somailer (v.0.2.6)[98]; (3) no obvious large copy number detected in normal samples by the Battenberg algorithm (suggesting no tumor and normal sample swapping[59]); (4) tumor exclusions if signature SBS7 (exposure to ultraviolet light, potential metastatic melanoma) or SBS31 (prior chemotherapy treatment with platinum drugs) are identified by mutational signature analysis; (5) Exclusion of samples with low-quality whole genome sequencing data if the number of detected somatic alterations < 1000 and NRPCC (number of reads per tumor chromosomal copy, representing the sequencing coverage per haploid genome) < 10, or if the number of somatic alterations < 100.

### Whole genome sequencing and data preprocessing
Frozen tumor samples and matched normal controls from 221 smokers with lung cancer from the EAGLE study[33] were selected for whole genome sequencing. DNA extraction and sequencing library construction followed our previously reported protocol[59]. Illumina HiSeq X was used for performing WGS at the Broad Institute following standard Illumina protocols with targeted sequencing depth 80x for tumor and 30x for normal tissue samples. The raw sequencing data in FASTQ format were collected, and the GATK best practices workflow was applied to generate the analysis-ready CRAM files using GRCh38 as the human reference genome. For the WGS data from public databases, the preprocessed aligned BAM/CRAM files (including unmapped reads) were first converted back to FASTQ files using Bazam (v.1.0.1)[99] to retain the sequencing lane and read group information and then processed using the same pipeline as for the EAGLE dataset.

### Genome-wide somatic alteration calling
Genome-wide somatic mutations were called from analysis-ready CRAM files following our previous bioinformatic pipelines[59]. Four different calling algorithms were applied using tumor-normal paired analysis, including Strelka (v.2.9.10), MuTect, MuTect2, and TNscope. Sentieon's genomics software (v202010.01) was used to run the MuTect (named TNsnv), MuTect2 (named TNhaplotyper), and TNscope algorithms. An ensemble method was applied to merge different callers with additional filtering to reduce false positive calling[59]. The final mutation calls for both SNVs and indels needed to meet the following criteria: (1) read depth > 12 in tumor and > 6 in normal samples; (2) variant allele frequency < 0.02 in normal samples, and (3) overall allele frequency (AF) < 0.001 in multiple genetics databases including 1000 Genomes (phase 3 v.5), ExAC (v.0.3.1), and gnomAD exomes (v2.1.1) and genomes (v3.0). For indel calling, only variants called by at least three algorithms were kept, followed by left normalization.

For somatic copy number alterations (SCNAs), the Battenberg algorithm (v.2.2.9)[100] was used to generate the initial SCNA profile, followed by assessment of clonality of each segmentation, as well as tumor purity and tumor ploidy. Battenberg refitting was repeated with a suggested tumor purity and ploidy as input until the final SCNA profile was retained (see below). Recurrent copy number alterations at the gene level were identified using GISTIC v.2.0[101] based on the major clonal copy number for each segmentation. Two algorithms, Meerkat (v.0.189)[102] and Manta (v.1.6.0)[103], were applied with recommended filtering for identifying structural variants (SVs), and the union set of these two callers was merged as the final SV dataset. A window of 50 bp was used when merging SVs. For the identification of putative transposon elements (TE), we used a pipeline called TraFiC-mem (v.1.1.0) (Transposome Finder in Cancer)[104], and only the TE insertions that passed the default filtering were kept. We define the overall somatic alterations as the summary of SNVs, indels, SVs, and TE insertions.

### Consensus estimation of tumor purity, ploidy, and clonality
For a consensus estimation of tumor purity, we systematically compared three approaches based on different genomic features as previously reported[59]: SCNA profiling using Battenberg; mutation clustering (VAF/CCF distribution) using Bayesian Dirichlet Process (DPClust v.2.2.8)[100,105]; and bayesian mixture models based on both SCNA and mutation clustering using CCUBE[106]. First, the SCNA calling was performed at the sample level based on the Battenberg algorithm, followed by the DPClust process. Overall genomic matrices were generated, including the purity estimation consistency among Battenberg, DPClust, and CCUBE; ploidy status; superclones or superclusters (CCF > 1); percentage of genomic alterations by SCNA (PGA); homozygous deletion size; percentage of segmentation with loss of heterozygosity, etc. Unsupervised hierarchical clustering was performed based on this matrix, which allows the identification of different groups of tumors with similar genomic profiling features and to flag samples for refitting (e.g., good profile, swapped diploid and tetraploid status, over/underestimated purity, incorrect segmentation for copy number alterations). Battenberg was then used to calculate new purity and ploidy values for refitting. This process was repeated on flagged tumor samples until a good genomic profile was retained with consensus tumor purity estimation.

### Mutational signature analysis and identification of the APOBEC mutator phenotype
SigProfilerMatrixGenerator[107] was used to create the mutational matrix for all types of somatic mutations. The deconvolution of mutational signatures was performed by SigProfilerExtractor (v.1.1.3)[28] using the WGS setting and the COSMIC mutational signatures as reference (v3.2). Hierarchical clustering of the contribution of mutational signatures was performed using Euclidean distance and Ward's minimum-variance clustering, separated by the presence of signature SBS4 and APOBEC signatures SBS2 and SBS13. SigProfilerClusters (v.1.0.0)[108] was used to identify *kataegis* and other clustered mutation events (e.g., *Omikli*). Mutational signature analyses were repeated based on the clustered mutations.

To further verify the APOBEC mutational loads, we also analyzed the pattern of mutagenesis by APOBEC cytidine deaminase using P-MACD[5,17]. A mutation annotation file (MAF) was used as input, and sample-specific summaries were generated, including "APOBEC_MutLoad_MinEstimate", which were compared against the number of APOBEC mutations (SBS2 + SBS13) estimated by SigProfileExtractor. In addition, we estimated the percentage of hypermutable strand-coordinated ssDNA (scssDNA) per genome, which can represent events of break-induced replication occurring at positions of double-strand breaks that cannot be repaired by canonical DSB repair[109]. Mutation clusters from P-MACD analysis with $P \leq 10^{-4}$ were considered bona fide mutation clusters. Mutation clusters with < 10 bp inter-mutational distances were considered as complex events and counted as one mutation, which originated from a single mutagenic event rather than from independent DNA lesions. Mutation clusters with more than 3 mutations containing only mutated Cs or only mutated Gs were used as a proxy for the long persistent regions of scssDNA. These clusters are associated with the enrichment of APOBEC mutation signatures in cancer[5,21,110,111]. The length of a hypermutable ssDNA region was estimated as the distance between mutations bordering a "C" or "G" coordinated cluster.

To distinguish the APOBEC hypermutation signature from APOBEC3B mutagenesis, we followed a previous procedure to calculate Y/RTCA enrichment and identify A3A-like or A3B-like tumors[17]. Given the null hypothesis of random mutagenesis, the expected number of YTCA mutations was calculated based on the fraction of motifs at YTCA from mutations within the TCA context. The expected number of RTCA and NTCA mutations ($N$ = any base) were computed analogously. The $\chi^2$ test for goodness of fit was used to identify samples that had a ratio of YTCA to RTCA mutations that differed statistically from random, by comparing observed *vs.* expected mutation counts. *P*-values were corrected by the Benjamini-Hochberg method, with Q < 0.05 considered significant.

## Estimation of the age at onset of the most recent common ancestor (MRCA) and tumor latency

We estimated the tumor chronological expansion, including the occurrence of the most recent common ancestor (MRCA) and the tumor latency (calculated as the difference between age at tumor diagnosis and age at the occurrence of the MRCA), using an approach from our previous studies[44,59]. For this analysis, we only included 151 high-quality tumors (tumor purity > 0.3 and NRPCC (representing the sequencing coverage per haploid genome) >10) with age at diagnosis information. In summary, the number of clock-like CpG > TpG mutations in a NpCpG context (SBS1) was counted for all tumors, and a hierarchical bayesian linear regression was fit to relate to age at diagnosis, accounting for tumor ploidy, tumor purity, and subclonal architecture. The age at the occurrence of the MRCA was estimated for each tumor, adopting an estimated tumor acceleration rate of 5x due to the high mutational burden as previously shown[44]. We subtracted the age at occurrence of the MRCA from the age at tumor diagnosis to estimate the tumor latency or subclonal diversification.

## Cell of origin (COO) inference from genome-wide mutational patterns

COO inference was performed using a recently developed procedure (manuscript in revision). Briefly, a comprehensive lung cell atlas was collected from several lung scRNA-Seq studies[112–115]. The standard approaches for scRNA-Seq data processing and integration were applied, identifying seven benign lung cell types: basal, neuroendocrine, goblet, ciliated, club, alveolar type I, and alveolar type II cells. Gamma-Poisson regression (SNV ~ cell type expression + covariates) was applied to correlate each cell type-specific gene expression profile with somatic SNV profiles from our study. This regression was adjusted for covariates, including replication timing, GC context, average gene expression across all cell-type-specific centroids, log intronic fractions, and log exonic fractions. A separate regression was fitted for each cell type-specific gene expression profile. Mitochondrial genes, as well as any genes with incomplete intronic or exonic data, were removed during our analysis. The maximum likelihood regression fit was used to obtain the main effect size (relative size) and 95% confidence interval on the relative risk, which represents the association between the tumor and one of seven lung cell types. To assess significant deviation of the relative risk from the baseline, we performed the Wald test to compute a two-tailed P-value. The lung cell type with the lowest relative risk in each tumor sample represents the putative cell of origin for that tumor. We compared the assigned cell of origin between LAS and HAS tumors and also compared the relative risk for specific cell types. In addition, lung cell-type-specific gene expression markers for AT2 and basal cells, collected from LUAD single-cell RNA-Seq studies[60] and the CellMarker2.0 database[116], were utilized to validate these findings.

## Neoantigen prediction

To predict neoantigens, patient-specific HLA haplotypes were identified using HLA-HD (v.1.2.1)[117]. The software NetMCHpan4.1[118] was then run on 9–11 neo peptides derived from all nonsynonymous mutations, taking into account the patient's specific HLA genotypes. NetMCHpan4.1 was included in the NeoPredPipe pipeline (v.1.1)[119]. The decomposed mutation probabilities generated by the SigProfilerExtractor were used to predict the neoantigens derived from specific mutational signatures. For TCGA data, the predicted SNV neoantigen counts per sample were collected from a previous publication[120]. We used the number of peptides predicted to bind with MHC proteins as the neoantigen load in our analysis.

## Genomic enrichment analyses between LAS and HAS tumors

We identified cancer driver genes using the intOGen pipeline[121], and 30 genes with a combined q-value < 0.1 were selected as the driver genes for enrichment. Logistic regression was performed using these

genomic alterations (nonsynonymous mutation status) as an outcome and tumor subtype as a variable, adjusting for the following covariates: age, sex, TMB, tumor purity, and histology. In addition, we further restricted the analyses to mutations predicted to be drivers based on the Cancer Genome Interpreter platform[122] (https://www.cancergenomeinterpreter.org/home). Specifically, the definition of "driver mutation" was based on a machine-learning approach (BoostDM) and a rule-based approach (OncodriveMUT). To be defined as a "driver" the mutation had to be identified as such by both BoostDM and OncodriveMUT.

## Smoking variables association analysis

To investigate associations between genomic data and tobacco smoking variables, we evaluated five variables, including CIGT_AGE_1-BEGIN = Age (in years) when subjects started smoking cigarettes regularly for the first time; CIGT_PER_DAY = Average intensity of cigarette smoking, measured as the number of cigarettes per day; CIGT_TIME_-LAST_QUIT = Number of years since the subject quitted smoking cigarettes (0 means current smokers); CIGT_TOT_DURATION = Total period (in years) during which the subject smoked cigarettes regularly; TTFC = Time to first cigarette in the morning (from the first question of the Fagerstrom test for nicotine dependence: 'How soon after you wake up do you smoke your first cigarette?'). We performed multivariate regression analyses to assess the associations between smoking variables and genomic features, including TMB, SBS4, DBS2, ID3, and the number of mutations for each mutation category. To evaluate the independent effects of smoking variables, we included five categorical smoking variables in the same model while adjusting for age, sex, histology, and tumor purity as covariates. Additionally, we conducted trend testing by treating smoking variables as continuous instead of categorical to assess overall trends. To control for multiple comparisons between LAS and HAS tumor groups, FDR-adjusted p-values were calculated for TTFC, which was the only smoking variable significantly associated with TMB.

## Methylation data analysis

Genome-wide DNA methylation was profiled on the Illumina HumanMethylationEPIC BeadChip (Illumina, San Diego, USA). Genomic DNA was extracted, and DNA methylation was measured according to Illumina's standard procedure at the Cancer Genomics Research Laboratory (CGR), National Cancer Institute. Raw DNA methylation data (".idat" files) were generated for our study, combined with methylation data collected from the public TCGA LUAD study (Illumina Human Methylation 450k BeadChip) (Supplementary Data 1), and processed using RnBeads (v.2.0)[123]. Background correction ("enmix.oob") and beta-mixture quantile normalization ("BMIQ") were applied. Unreliable probes (Greedycut algorithm with detection P < 0.05), cross-reactive probes, and probes mapping to sex chromosomes were removed. Samples with outlier intensities in 450k/EPIC array control probes were removed from the dataset as described in the RnBeads vignette. We performed principal component analysis (PCA) of these samples based on the genotyping probes and removed subjects with Euclidean distance between matched tumor/normal pair > upper quartile + 3*inter-quartile range (IQR).

We implemented batch effect correction using the Combat algorithm, applied separately to tumor and normal samples. This was done using the "sva" R package[124] on M-values, accounting for known technical factors such as study cohort, collection sites, sample plates, sample wells, array types (450 K and EPIC), chip ID, and positional effects. To evaluate the presence of any residual batch effects, we utilized the MBatch tool (v2.0, available at https://github.com/MD-Anderson-Bioinformatics/BatchEffectsPackage). The analysis yielded Dispersion Separability Criteria (DSC) values of < 0.5 for all technical factors, indicating that batch effects were minimal following correction. The batch-adjusted methylation-level beta values were then employed in subsequent association analyses.

We selected 952 probes that had genome-wide significant associations with smoking variables in a recent study[58]. The P-value and methylation status (hypermethylation vs. hypomethylation) of these probes were included in our visualization of the association results. Linear regression was performed for each smoking variable using the methylation beta value as an outcome and adjusting for the following covariates: age, sex, histology, and tumor purity for the tumor methylation analysis, and age and sex for the methylation analysis of normal tissue. Association P-values were further corrected using the Benjamini-Hochberg method. These analyses were then repeated after separating LAS and HAS tumors. In addition, multivariate regression analyses were also performed, including both CIGT_TIME_LAST_QUIT and CIGT_TOT_DURATION as well as age and sex as covariates in the methylation analysis of normal tissue.

In addition, EpiTOC2[125] were used to estimate the intrinsic stem cell division rate or mitotic clock based on the methylation status of 163 EpiTOC2 CpG probes.

### RNA-Seq data analysis

RNA-Seq data from the EAGLE samples were generated using the Illumina HiSeq platform and Illumina TruSeq Stranded Total RNA-Seq protocol, producing 2 × 100bp paired-end reads. The RNA-Seq FASTQ files from the EAGLE study we used for this analysis are also available in dbGaP under the accession number phs002346.v1.p1[126]. The RNA-Seq FASTQ files from the TCGA-LUAD study were downloaded from the GDC Data Portal[127]. The FASTQ files were aligned to the human reference genome GRCh38/hg38 using STAR[128], annotated using the GENCODE v35, and processed by HTSeq[129] for gene quantification. The resulting expression data were corrected for batch effects using the R package ComBat-seq[130] and normalized to Counts Per Million (CPM), followed by log2 transformation using the R package edgeR[131]. Only 'expressed' genes defined by CPM > 0.1 in at least 10% QC-passed samples were included in the final quantification data.

We performed multivariate analyses to identify genes whose expression was associated with TTFC after adjusting for tumor purity and copy number status. Regression analyses were performed separately within the LAS and HAS subtypes. Genes with an association $P < 0.05$ were selected for pathway analyses using Ingenuity Pathway Analysis (IPA). In addition, we performed differentially expressed gene (DEG) analyzes between LAS and HAS tumors based on the two-sided Wilcoxon rank-sum test. The batch effect-corrected read counts were used as input for SARTools (v.1.8.0)[132], and edgeR (v3.36)[131] was used to identify the DEGs. The top 500 significant DEGs within each comparison were selected for pathway analyses by IPA. Significant pathways were selected based on |Z-score| > 1 and FDR < 0.05. Gene Set Enrichment Analysis (GSEA) was conducted using the clusterProfiler package in R[133], based on the hallmark gene sets from the Molecular Signatures Database (MSigDB) (https://www.gsea-msigdb.org/).

For immune cell type decomposition using normalized gene expression data from each patient, the xCell tool[134] was utilized to estimate the abundance of various immune cells within tumor tissues. Comparisons of each cell type between LAS and HAS tumors were conducted using the two-sided Wilcoxon rank-sum test. Immune cells with an overall proportion greater than 0.1 were included in the analysis. Differences were considered statistically significant if the false discovery rate (FDR) adjusted p-value was less than 0.05.

### Enrichment and association validation using TCGA and PCAWG data

For TCGA PanCancer data, we downloaded the genomic alterations, gene expression, and clinical information from the cBioPortal for Cancer Genomics (https://www.cbioportal.org/datasets; study: TCGA PanCancer Atlas). For PCAWG PanCancer data, the genomic and clinical information was collected from the ICGC Data Portal (https://dcc.icgc.org/pcawg). The mutational signature decomposition data for both TCGA and PCAWG studies were extracted from a previous publication[3] based on the SigProfilerExtractor algorithm. A minimum of 50 mutations assigned to SBS2 + SBS13 was required for the identification of HAS tumors. Tumor purity was derived from a consensus measurement of purity estimates (CPE) from a previous study[135]. Logistic regression was used for genomic features enrichment analyses between LAS and HAS tumors, adjusting for age, sex, TMB, and tumor purity. Two-sided Wilcoxon rank-sum test was used to compare the age of diagnosis between LAS and HAS, with at least 5 patients for each group. In addition, DNA methylation and tobacco smoking data from TCGA LUAD were downloaded from the Genomic Data Commons (GDC) data portal (https://portal.gdc.cancer.gov/).

### Data analysis of the independent LUAD WGS dataset from TCGA

For the validation dataset, we collected BAM files of newly sequenced WGS data from TCGA LUAD in smokers (GDC Data Portal, https://portal.gdc.cancer.gov/). We applied the same Sherlock-*Lung* bioinformatic pipelines for data quality control, genomic alteration calling, and downstream analyses as previously described. We excluded any TCGA samples that were part of the discovery dataset, and also excluded tumor samples with fewer than 100 identified genomic alterations due to lower tumor purity. Additionally, we collected clinical and exposure information from the GDC Data Portal. Similar to the TCGA study, we calculated pack years smoked to represent lifetime tobacco exposure, defined as the number of cigarettes smoked per day multiplied by the number of years smoked, divided by 20. We also collected both tumor and normal RNA-Seq data from the same TCGA LUAD tumors from the cBioPortal, quantified as RNA Seq V2 RSEM.

### Interplay between APOBEC activity and tobacco-smoking induced mutagenesis

To assess whether the associations between APOBEC subtypes and driver mutations in *TP53* depend on smoking-related variables, we used TTFC as the primary smoking variable. TTFC was selected as it showed the strongest associations with genomic features in our preliminary analyses[46]. We performed logistic regression analyses using the generalized linear model (GLM) framework in R, with the following model:

$$\text{glm(Driver Gene Mutation Status} \sim \text{APOBEC\_Subtype*TTFC}$$
$$+ \text{Age} + \text{Sex} + \text{Histology} + \text{Tumor\_Purity, family} = \text{"binomial")}$$

Significance of the interaction term was assessed to determine whether the smoking variable TTFC modifies the relationship between APOBEC mutagenesis and specific driver gene mutations in *TP53*. Our analysis revealed a significant interaction between TTFC and APOBEC subtypes ($P_{\text{interaction}} = 0.046$). This finding indicates that smoking intensity, as measured by TTFC, influences the association between APOBEC mutagenesis and *TP53* mutations.

### General statistical analyses

All statistical analyses were performed using the R software v4.1.2 (https://www.r-project.org/). In general, the Mann-Whitney (two-sided Wilcoxon rank-sum) test was used for two-group comparisons, and the two-sided Fisher exact test was applied for the enrichment analysis of two categorical variables. $P < 0.05$ was considered statistically significant. If multiple testing was required, we applied the FDR correction based on the Benjamini–Hochberg method. For the survival analyses, a proportional-hazards model was used to investigate the associations between APOBEC subtypes and overall survival, adjusting for age, sex, stage, and histology.

### Reporting summary

Further information on research design is available in the Nature Portfolio Reporting Summary linked to this article.

## Data availability

WGS raw data (CRAM files) and methylation raw data (intensity idat files) used for this work have been deposited in the dbGaP under accession number phs002992.v1.p1. The RNA-Seq raw data (FASTQ files) from EAGLE subjects can be accessed through dbGaP with the accession number phs002346.v1.p1. The access information for the public multi-omics datasets can be found in Supplementary Data 1. Independent raw WGS data from the TCGA LUAD study are available through the Genomic Data Commons (GDC) data portal (https://portal.gdc.cancer.gov/) under TCGA data access policies. Source data are provided with this paper. Access Conditions: Raw data access is subject to approval by an NCI Data Access Committee (DAC) upon submission of a data access request through the dbGaP system. Access Limitations: Only bona fide investigators may be granted access to these data based on DAC approval and compliance with data use limitations (DULs). DULs for the data in this study include research related to lung cancer or smoking. Contact Information: For questions regarding data access, users can contact the NCI Office of Data Sharing (NCIOfficeofDataSharing@mail.nih.gov). Estimated Timescale: Access decisions are typically made within a few weeks of request submission. Source data are provided in this paper.

## Code availability

The major bioinformatics pipelines, including genome-wide somatic callings based on WGS data, can be found at https://github.com/xtmgah/Sherlock-Lung. Battenberg SCNA calling algorithm can be found at https://github.com/Wedge-lab/battenberg. Dirichlet process-based methods for the subclonal reconstruction of tumors can be found at https://github.com/Wedge-lab/dpclust. Mutational signature analysis, SigProfilerExtractor, can be found at https://github.com/AlexandrovLab/SigProfilerExtractor. The code for P-MACD can be found at https://github.com/NIEHS/P-MACD. The algorithm for the identification of clustered mutations and events is available at https://github.com/AlexandrovLab/SigProfilerClusters.

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

## Acknowledgements

This work has been supported by the Intramural Research Program of
the Division of Cancer Epidemiology and Genetics (project ZIACP101231
to M.T.L.), National Cancer Institute, and by the Intramural Research
Program of the National Institute of Environmental Health Sciences
(project Z1AES103266 to D.A.G.), National Institutes of Health (NIH).
Cancer research in the Harris lab is supported by NCI P01-CA234228,
NCI P50-CA247749, and a CPRIT Established Investigator Recruitment
Award. RSH is an Investigator of the Howard Hughes Medical Institute
and the Ewing Halsell President's Council Distinguished Chair at the
University of Texas Health San Antonio. Research performed for this
publication at the Alexandrov Lab was supported by a Packard Fellow-
ship for Science and Engineering as well as by grants from the US
National Institutes of Health, including: NIEHS R01ES030993-01A1,
NIEHS R01ES032547, and NCI R01CA269919. We thank Cameron Durfee
for helpful comments. This work utilized the computational resources of
the NIH HPC Biowulf cluster (https://hpc.nih.gov).

## Author contributions

M.T.L. and T.Z. conceived and designed the study, with contributions
from J.Sa., T.Z., J.Sa., P.H.H., W.Z., K.E.J., L.J.K., J.M., A.K., C.W., E.N.B.,
M.D.G. and R.V. performed formal data analyses. T.Z., B.Z., T.M.P., J.Sh.,
M.I., D.C.W., D.A.G., L.B.A., R.S.H. and M.T.L. developed methodology.
T.Z., J.Sa, P.H.H., J.R., F.C.-M., A.H., S.M.L. and N.C. curated samples and
data. A.C.P., D.C., S.J.C. and M.T.L. provided resources. R.H. reviewed
tumor histology features. T.Z and J.Sa. performed data visualization and
validation. T.Z., J.Sa. and M.T.L. wrote the original draft. B.Z., J.Sh.,
D.C.W., D.A.G., S.J.C., L.B.A. and R.S.H. reviewed and edited the
manuscript. M.T.L. supervised the project. All authors read and
approved the final manuscript.

## Competing interests

L.B.A. is a compensated consultant and has equity interest in io9, LLC.
His spouse is an employee of Biotheranostics, Inc. L.B.A. is also an
inventor of a US Patent 10,776,718 for source identification by non-
negative matrix factorization. E.N.B. and L.B.A. declare U.S. provisional
patent applications with serial numbers 63/289,601 and 63/269,033.
L.B.A. also declares U.S. provisional patent applications with serial
numbers: 63/366,392; 63/367,846; and 63/412,835. All other authors
declare no competing interests.

## Additional information

**Supplementary information** The online version contains
supplementary material available at

Maria Teresa Landi.

**Peer review information** *Nature Communications* thanks the anon-
ymous reviewer(s) for their contribution to the peer review of this work. A
peer review file is available.

[1]Division of Cancer Epidemiology and Genetics, National Cancer Institute, Bethesda, MD, USA. [2]Westat, Rockville, MD, USA. [3]New York Genome Center, New
York, USA. [4]Integrative Bioinformatics Support Group, National Institute of Environmental Health Sciences, Research Triangle Park, NC, USA. [5]Manchester
Cancer Research Centre, The University of Manchester, Manchester, UK. [6]Department of Cellular and Molecular Medicine and Department of Bioengineering
and Moores Cancer Center, University of California San Diego, La Jolla, CA, USA. [7]Cancer Genomics Research Laboratory, Frederick National Laboratory for
Cancer Research, Frederick, MD, USA. [8]National Center for Biotechnology Information, National Library of Medicine, National Institutes of Health, Bethesda,
MD, USA. [9]Department of Pathology, Yale School of Medicine, New Haven, CT, USA. [10]Department of Clinical Sciences and Community Health, University of
Milan, Milan, Italy. [11]Fondazione IRCCS Ca' Granda Ospedale Maggiore Policlinico, Milan, Italy. [12]Genome Integrity and Structural Biology Laboratory, National
Institute of Environmental Health Sciences, Research Triangle Park, NC, USA. [13]Department of Biochemistry and Structural Biology, University of Texas Health
San Antonio, San Antonio, TX, USA. [14]Howard Hughes Medical Institute, University of Texas Health San Antonio, San Antonio, TX, USA. [15]These authors
contributed equally: Tongwu Zhang, Jian Sang. ✉e-mail: landim@mail.nih.gov

