## [Transparent Peer Review file · Nature Communications]

APOBEC affects tumor evolution and age at onset of lung cancer in smokers

Corresponding Author: Dr Maria Teresa Landi

Version 0:

Reviewer comments:

Reviewer #1

(Remarks to the Author)

This is an extremely accurate and important study where the authors present a multi-omics analysis including newly sequenced deep whole-genome, transcriptomic, and epigenomic profiles of 309 paired tumor-normal lung tissue samples from smokers.

They suggest that APOBEC mutagenesis operates in all cancer groups, but at different levels, a “Low APOBEC Subtype” (LAS) and a “High APOBEC Subtype” (HAS). No significant differences were observed between LAS and HAS in terms of potential confounders. However, HAS tumors exhibited a significantly higher number of retrotransposon insertions, as expected for a subtype with higher genomic instability.

Smoking was associated with genomic changes only in LAS tumors. Also, tobacco smoking-induced epigenomic changes were reversed after quitting smoking only in LAS tumors: the association between cg05575921 (AHRR) methylation and time since quitting smoking was observed only in LAS. In contrast, genes whose methylation levels are not associated with tobacco smoking showed no difference between LAS and HAS.

There are many other interesting observations that are relevant to the clinical management of lung cancer: methylation based mitotic rate in HAS tumors showed lower cell division than LAS tumors, and patients with HAS tumors showed a later age at diagnosis. Overall, the findings suggest that APOBEC mutagenesis has a stronger effect on the tumor clonal expansion, i.e. when tobacco smoking mutagenesis has its strongest effect.

In brief, a carefully conducted, well-described study with important findings.

I only have two question: is there any chance of replicating the findings in another cohort? They do not address the problem of subgroup analyses, that are many (any statistical correction/appraisal for this?).

Reviewer #2

(Remarks to the Author)

The study by Zhang et al performed multiomic profiling of 309 paired lung tumor-normal tissues, mostly lung adenocarcinomas (LUADs). By mutational signature analysis, the group found samples were divided mainly into two groups with low (LAS) and high (HAS) APOBEC mutagenesis. They found that LAS were enriched with A3B-like mutagenesis, KRAS mutations, higher mitotic rate, AT2/AT2-like expression and distal disease based on inference of cells-of-origin (COO), younger age of onset. In contrast, the HAS group were enriched with A3A-mutagenesis, TP53 mutations, lower mitotic rate, basal expression and proximal disease based on COO, and older age of onset. LAS had higher association with DNA repair (UNG, BER). LAS more likely showed associations between time to first cigarette (TTFC) and mutation burden as well as between time from quitting smoking and methylation (e.g., in AHRR) changes. HAS showed more predicted neoantigens than LAS. Given the above, the group construes that higher mutation burdens with lower DNA repair in HAS leads to senescence which then leads to mitotic arrest in presence TP53 mutations. This, along with the inferred lung regeneration occurring in HAS leads to a longer time to accumulation of mutations which, along with increased neoantigens that may indicate increased immune surveillance, explain the longer time to lung tumor onset. While the study represents a

significant effort in multiomic profiling, this reviewer has major concerns with the speculative nature of the conclusions and some aspects in the study's design and presentation. The following comments need to be addressed prior to publication for the astute readership of Nature Communications.

-The study finds that LAS is more enriched with KRAS mutations, and HAS tumors are more enriched with TP53 mutations. While this appears to be the case (and the study postulates that these genomic differences are associated with differing genomic instability, cells of origin, among other variables), there are still quite a bit of TP53 mutations in LAS and KRAS variants in LAS. It is not clear then whether the paper suggests that KRAS mutant tumors in LAS and HAS are different (?). Another missed opportunity, perhaps concern here, is that TP53 and KRAS mutations typically co-occur (KP) and it has been shown that KP and KRAS mutant lung adenocarcinomas with wild type TP53 are almost completely different tumors. In this theme, is it not clear whether the biology of KP tumors is accounted for in the study's analysis and hypothetical conclusions.

-There are various differences in measures between groups accentuated by the study that are at best modest, such as the association between short TTFC and mutated genes associated with smoking (ED Fig. 7), expression of KRT15, TP63 between LAS and HAS (Fig. 4a), etc.

-This reviewer has major concerns in regards to the speculative conclusions that are drawn from analyses that are best correlative (although adequate). Such as: Strong DNA damage induced by APOBEC3A hypermutation lead to cell regeneration. HAS tumor composition likely includes a higher number of newly generated or de-differentiated cells, which do not display the differentiated LAS tumors."

-On that theme, this reviewer has major concerns with the COO analysis which is based on a method by the group currently under peer review and there are insufficient details in the Methods with regards to this analysis. The study suggests that Club, basal, ciliated cells are enriched in HAS whereas distal AT2 cell types are enriched in LAS. Review of z-scores in Supp. Fig 18 suggests that the pulmonary healing signaling pathway (how many genes are in this pathway? and what are the genes?) is activated in both LAS and HAS. It is also difficult to make sense of Supp. Fig. 19 since most tumors in LAS and HAS show an ambiguous COO and the associations in the heatmap in Supp. Fig. 19a (not sure if the association scale is correct) are not clear. The discussion section mentions that LAS is enriched with KRAS mutations and that AT2 cells are likely the cells of origin of KRAS mutant tumors. While this may be true, many reports have shown that KRAS mutant LUADs have reduced alveolar differentiation and in fact lose AT2 markers (e.g., mucinous KRAS-mutant LUADs). Additionally, various studies have shown that KRAS mutations in AT2 cells are drivers of regeneration in alveolar niches. These previously published reports are contradictory to the conclusions (e.g. LAS low regeneration more KRAS, HAS high regeneration less KRAS) of the study. Since the study centers on cohorts that are richly annotated, could the group review the clinicopathological data to determine anatomical (distal versus proximal) of LUADs in the LAS and HAS groups?

-The study construes that HAS tumors, given the higher neoantigens, could less likely evade host immunity which could partially explain the longer time (higher age of patients) lung tumor onset. This is a conclusion that while being interesting is also speculative. Could the group deconvolute the RNA-seq data to determine whether HAS tumors have higher immune infiltration compared with LAS tumors?

-PD-L1 RNA is modestly different between LAS and HAS. Could PD-L1 expression differences be confirmed in a subset of LAS and HAS cases using standard (and more appropriate for PD-L1) immunohistochemistry?

-The presentation of the study is not adequate, there are very few panels/data in the main figures (are these the only main talking points?) and the data in the ED and supp figures are too sparse and could easily be consolidated.

Reviewer #3

(Remarks to the Author)

This study presents a multi-omics characterization including WGS of hundreds of lung tumors newly processed and data generated. Patterns of APOBEC mutagenesis are analysed in context of smoking phenotypes and mutational signatures, and in context of gene expression patterns and DNA methylation patterns.

Regarding the association of high-APOBEC or low-APOBEC mutagenesis tumors with mutations in certain driver genes (here, TP53 and KRAS) – associations of signatures with driver events were addressed in various prior studies e.g. PMID: 30412573, PMID: 29748584. Implications of these TP53 and KRAS associations with APOBEC mutagenesis to cancer evolution are not clear. Similarly, concluding that APOBEC may have caused some of these mutations just based on the context enrichment is perhaps premature.

Regarding the association of UNG expression levels with APOBEC3B (but not 3A) expression; it was not explored how this association bears upon APOBEC mutagenesis levels – does the UNG actually prevent mutagenesis? The statement „This finding suggests that UNG expression is activated by uracils in DNA generated by APOBEC3B mutagenesis or that UNG and APOBEC3B share common regulatory mechanisms^{43,44}.“ is highly speculative and joins the plethora of other heavy speculation sprinkled through the Results section. The coexpression analysis gives no indication of direction of effect or confounding. Attempting to control for subtype and impurities would help make this result firmer. Also, it is not clear that the association of $P=8e-4$ would stand after a FDR correction – I presume that many genes were tested. I find the logic in the clonality commentary is not well explained; it is not clear how clonality supports episodic activity of APOBEC or lack thereof, nor is episodic activity itself sufficient reason for lack of correlation to gene expression.

Regarding „as well as a higher proportion of APOBEC mutations contributing to kataegis (61.4% versus 50.8%; Fig. 1d-e), even in clusters of very low numbers of mutations (41.7%).“ Are the clusters with low numbers of mutations omikli? They would need to mention them as such (they do so in the methods).

The section „Tobacco smoking addiction is associated with genomic changes only in LAS tumors“ which appears central to the story (suggesting interaction between smoking and APOBEC signatures) has some premature conclusions and speculation. For example, why would the “time to first cigarette (TTFC) in the morning, a marker of strong nicotine addiction“ be a better measure than more direct measures of tobacco exposure seems odd (their explanation of better DNA repair in the morning seems a bit of a stretch). Next, in the high-APOBEC tumors they do not find an association between smoking exposure and smoking signature, which they explain by a non-parsimonius mechanism (lines 208-212), involving a lot of steps for which there appears not to be supporting data. Some of the steps do not even appear plausible at an initial inspection, such as TP53 mutant cells having more apoptosis (mutation in TP53 should reduce ability to apoptose not vice versa). The general idea is that APOBEC3A activity kills cells, and for some reason preferentially clearing the tobacco smoke mutant ones. Unfortunately this is not the most convincing in the absence of more specific data supporting it.

The DNA methylation analysis in the lines 213 – 237 is interpreted such that lack of association between certain probes and smoking phenotypes in the high-APOBEC tumors somehow suggests changes in cell turnover, however the lack of association could result from a various technical and biological sources of noise.

That APOBEC (or whatever mutagen) generates more neoantigen is somewhat unsurprising given that APOBEC mutagenesis is known to be enriched in early-replicating, genic regions, compared to many other mutagens (esp. tobacco smoke) where mutations are depleted in early-replicating DNA.

Association of high-APOBEC mutagenesis with higher age is potentially interesting – did this replicate in the TCGA (or another cohort)? About their claim „These findings suggest that APOBEC mutagenesis has a stronger effect on the tumor clonal expansion [...] than on the subsequent tumor progression.“ it is not clear how this claim of clonality connects to the age-analysis section; maybe it is a reference to the VAF analysis (lines 180-189).

Typo „inclined“.

Version 1:

Reviewer comments:

Reviewer #1

(Remarks to the Author)

I have no additional comment. The authors have responded adequately to my queries and suggestions.

Reviewer #3

(Remarks to the Author)

In the revised study, the authors have included a replication cohort, and reported several additional findings. There is some novelty in association of high-APOBEC tumors ("HAS") with age, however for some other findings reported, they instead confirm existing work. For instance, regarding link between A3 mutagenesis and immune infiltration added in revision: this mirrors prior reports of links between APOBEC expression and/or mutagenesis with tumor immunogenicity as reported for lung cancer and other cancer types (e.g. PMID:29695832, PMID: 31222843, PMID: 37215984). Next, links between A3 mutagenesis and mutations in PIK3CA and TP53 drivers were reported here as well as previously (PIK3CA in PMID: 35013316, PMID: 30412573; TP53 in breast and head&neck cancer in PMID: 37922356). Further findings related with association of high-APOBEC tumor genomes (here "HAS") with A3A and/or A3B gene expression levels are indeed numerous in the prior literature. The reported positive association of A3A-versus-A3B signature balance with overall A3 mutation burden ("HAS tumors were dominated (80.4%) by the A3A-like mutator phenotype, whereas LAS tumors were dominated (51.6%) by the A3B-like phenotype") mirrors the previously reported findings (PMID: 26258849). An important issue for authors to seriously consider is presenting various findings as novel either explicitly or having context suggesting that, while they were reported before.

Further, there remain issues to be sorted out regarding statistical support of findings, which presents, essentially, anecdotal examples in place of rigorous, systematic tests. This is of course convenient for getting passable p-values/FDRs however they should consider a more rigorous approach. Examples thereof would be as follows: (i) "HAS tumors exhibited... higher expression of basal cell markers (KRT15 and TP63)." and "LAS tumors... showed significantly higher expression of AT2 cell markers (NKX2-1 and SFTPB)." Which other genes have a stronger association than these hand-picked markers? (ii) "We observed a significant association between APOBEC3B and UNG and no association for APOBEC3A and UNG" How many other DNA repair or related genes have a stronger association with A3B mutagenesis (or A3A/B expression) than UNG does? The choice of UNG is based on some prior knowledge but many other genes could have overlapping mechanisms in the BER or MMR pathways. (ii-b) Related with this, another concern is that UNG association is FDR-corrected only over the APOBEC paralogs (n=10) however this may mislead because the association test is not to distinguish

one APOBEC paralog from another but rather one DNA repair gene from another, and DNA repair genes are >>10. (iii) There is implication that statistical interactions between genetics and smoking variables have been tested when this seems not to be the case "Other studies [...] have not specifically explored such associations in the context of a strong exposure to tobacco smoking.... Our study [...] investigating the association between APOBEC mutagenesis and driver gene mutations in the context of tobacco-smoking signatures." It is not clear to me that there is a statistical test showing that the APOBEC-TP53 or APOBEC-PIK3CA positive association is significantly different depending on smoking status (or on the smoking signature SBS4/SBS92), and similarly so for the APOBEC-KRAS negative association. There would need to be a test for interaction between smoking status or intensity, which does not seem performed, to be able to claim "Indeed, we found that the interplay between the APOBEC and tobacco smoking mutagenesis has significant implications for tumor evolution".

Finally, the presentation in the manuscript at various points can be improved:

- heavy speculation "Moreover, circadian disruption has previously been implicated in lung cancer tumorigenesis. Both catecholamine and glucocorticoid secretions are regulated by the circadian rhythm, and evidence suggests that their release is accelerated by components of cigarette smoke. When nicotine binds to nicotinic acetylcholine receptors expressed on adrenal medulla cells, it triggers the release of catecholamines into the bloodstream, which may promote epithelial cell proliferation, contributing to cell malignant transformation."

- generic statements, devoid of specific or convincing information "every analysis we have conducted follows logical steps to verify our hypothesis and explain the results. We used multiple approaches and data types to prove our hypothesis. This study is based on correlation but the large number of consistent results across approaches substantiate our claim."

- unclear implications of TP53 mutations on their model

... "We do not emphasize the role of TP53 in reducing or activating apoptosis directly in our model, although several papers do show that p53 mutants retain partial function and can induce apoptosis under certain conditions (PMID: 16543937; PMID: 9841917; PMID: 11790556; PMID: 11058599)." TP53 mutants may retain partial activity, however certainly less so than the wild-type TP53 (or, they can have a dominant-negative function thus additionally poisoning also the wild-type allele).

..."Mechanistically, single-cell transcriptomics analyses in mice models have shown that during LUAD evolution, p53 promotes AT1 differentiation through action in a transitional cell state. Notably, p53 also directs alveolar regeneration after injury by regulating AT2 cell self-renewal and promoting transitional cell differentiation into AT1 cells. Thus, p53 governs alveolar differentiation (PMID: 37468633) and this can contribute to the higher stem-like phenotype of the HAS (TP53-enriched) tumors." If functional p53 is supposed to contribute to higher stem-like phenotype -- as based on their literature survey -- this does not seem to fit with that they observe HAS to have mutated (dysfunctional) TP53? HAS cannot be said to be TP53-enriched if they are TP53 mutant, in fact just the opposite is true. This may be an inconsistency in their logic, or simply a wording issue that could be resolved by rephrasing.

- "Instead, we suggest that hypermutation by A3A and tobacco smoking mutagenesis, combined with limited DNA repair capacity and TP53-induced genomic instability, can trigger senescence, apoptosis, and cell regeneration" It was not clear to me where was the evidence for limited DNA repair capacity? Additionally, if anything, the A3A/A3B hypermutation has been linked to higher DNA repair activity by the HR, and also BER and MMR pathways in multiple studies.

- "the most significant probe (AHRH cg05575921) we identified is the very well-known CpG site associated with smoking cessation (PMID: 28100713). Thus, we concluded that these findings reflect true biological differences, such as increased cell turnover" It does not seem immediately clear how a significant CpG probe known to be associated with smoking cessation implies there is increased cell turnover.

Reviewer #4

(Remarks to the Author)

Zhang et al provide a description of a large novel WGS dataset and an interesting analysis related to differences between lung tumors with low or high APOBEC activity. They have substantially revised the manuscript in general and addressed the concerns of the previous reviewers. Overall, this is an impactful study that warrants publication in Nature Communications. I only have a small number of additional points and questions.

Major points:

There are some discrepancies between SigProfiler/NMF and P-MACD results for the APOBEC activity which adds some confusion. It appears that the LAS/HAS calls are derived from the SigProfiler results based on Figure 1A. Additional APOBEC mutations were identified in the LAS with the more sensitive P-MACD method which is specifically tailored towards characterization of APOBEC signatures. However, based on the P-MACD results in Supplementary Figure 3C,D, it is not clear that there are indeed two distinct groups of APOBEC samples. I suspect that the reason to keep the P-MACD results are to show that the LAS tumors were enriched in A3B-like mutations. But then it calls into question the overall grouping because some of the tumors in the LAS group have a higher number of APOBEC related mutations compared to some other tumors in the HAS group. While there is a significant shift in the mean, there is quite substantial overlap. If the LAS versus HAS grouping is made using the more sensitive P-MACD method, do the remaining results of the manuscript largely hold true?

The more "proximal-like" tumors from the COO analyses may refer to the Terminal Respiratory Unit (TRU) expression subtype of LUAD which were described in the TCGA marker paper (PMID: 25079552). This was formerly called bronchioid when first characterized (PMID: 17075127). This subtype tends to be enriched in EGFR rather than KRAS mutations.

Minor points:

I found this line in the abstract to be confusing because the inclusion of both previous literature and current findings:

“Hypermutation by unrepaired A3A and tobacco smoking mutagenesis combined with TP53-induced genomic instability can trigger senescence⁷, apoptosis⁸, and cell regeneration⁹, as indicated by high expression of pulmonary healing signaling pathway, stemness markers and distal cell-of origin in HAS.” Maybe a clearer way to state this is something like “Previous studies have shown that hypermutation by unrepaired A3A and tobacco smoking mutagenesis combined with TP53-induced genomic instability can trigger senescence⁷, apoptosis⁸, and cell regeneration⁹. We observed that HAS exhibited high expression of pulmonary healing signaling pathway, stemness markers and distal cell-of origin.”

Also in the abstract, the phrase “newly generated, unmutated cells” is not standard and not used in the manuscript. It is confusing because “unmutated cells” sounds like it is referring to normal cells but the results and conclusions are discussing cancer cells. Can you rephrase?

Version 2:

Reviewer comments:

Reviewer #4

(Remarks to the Author)

The authors have addressed by concerns and questions

Reviewer #5

(Remarks to the Author)

-

Version 3:

Reviewer comments:

Reviewer #5

(Remarks to the Author)

I thank the authors for addressing my all my comments. I have no further questions.

GENERAL RESPONSE TO ALL REVIEWERS

We appreciate the opportunity to revise our manuscript “APOBEC shapes tumor evolution and age at onset of lung cancer in smokers” based on the insightful comments from the reviewers. We have made substantial revisions to address each of the concerns raised. We have incorporated validation analyses using newly released TCGA LUAD whole-genome sequencing data. This independent dataset of 184 tumor samples allowed us to replicate our key findings, including the distinction between HAS and LAS tumors regarding APOBEC mutagenesis, retrotransposon insertions, driver mutations, APOBEC3A/B expression, basal and AT2 cell marker expression and age at diagnosis. These results enhance the robustness and generalizability of our study. We also performed additional statistical analyses, including controlling for multiple comparisons using the Benjamini-Hochberg procedure to address potential false discovery rates. We have clarified and explicitly stated these methods throughout our revised manuscript to ensure transparency and rigor. In light of the comments, we have carefully revised our conclusions. We clearly distinguish between data-driven findings and hypotheses that require further validation. This distinction is emphasized in the revised Discussion section. We have expanded the Methods section to provide comprehensive details on our analytical approaches, including cell-of-origin analysis and RNA-seq deconvolution for immune cell infiltration. These additions aim to enhance reproducibility and clarity for the readers.

(Please note, for convenience, all the figure numbers in the response below refer to the new figure numbers in the revised manuscript.)

SPECIFIC REVIEWER COMMENTS

Reviewer #1 (Remarks to the Author): expertise in lung cancer risk prediction and epidemiology

This is an **extremely accurate and important study** where the authors present a multi-omics analysis including newly sequenced deep whole-genome, transcriptomic, and epigenomic profiles of 309 paired tumor-normal lung tissue samples from smokers.

They suggest that APOBEC mutagenesis operates in all cancer groups, but at different levels, a “Low APOBEC Subtype” (LAS) and a “High APOBEC Subtype” (HAS). No significant differences were

observed between LAS and HAS in terms of potential confounders. However, HAS tumors exhibited a significantly higher number of retrotransposon insertions, as expected for a subtype with higher genomic instability.

Smoking was associated with genomic changes only in LAS tumors. Also, tobacco smoking-induced epigenomic changes were reversed after quitting smoking only in LAS tumors: the association between cg05575921 (AHRR) methylation and time since quitting smoking was observed only in LAS. In contrast, genes whose methylation levels are not associated with tobacco smoking showed no difference between LAS and HAS.

There are many other interesting observations that are relevant to the clinical management of lung cancer: methylation based mitotic rate in HAS tumors showed lower cell division than LAS tumors, and patients with HAS tumors showed a later age at diagnosis. Overall, the findings suggest that APOBEC mutagenesis has a stronger effect on the tumor clonal expansion, i.e. when tobacco smoking mutagenesis has its strongest effect.

In brief, a carefully conducted, well-described study with important findings.

I only have two questions: is there any chance of replicating the findings in another cohort? They do not address the problem of subgroup analyses, that are many (any statistical correction/appraisal for this?).

Response:

We thank the reviewer for the thorough review and insightful comments on our manuscript. We appreciate the recognition of the importance and accuracy of our study. We address the reviewer's specific concerns below:

Replication of Findings in Another Cohort:

We acknowledge the importance of replicating our findings in an independent cohort to confirm their robustness and generalizability. At the time of our initial data analysis, we collected all available deep whole-genome sequencing datasets of lung adenocarcinoma (LUAD) in smokers, including our EAGLE study and other published datasets, resulting in a total of 309 samples after quality control (QC). Recently, in 2024 TCGA released additional WGS data for LUAD. We collected these raw data files (BAM format) and applied the same bioinformatic pipelines as described in our manuscript. After strict QC, we identified 184 tumor samples as independent samples for the "validation" dataset (**Extended Data Fig. 8**).

We defined our original dataset as the “discovery” dataset. Using the same approach to define two subtypes, we observed that the validation dataset had a significantly lower proportion of HAS tumors (P=0.004; **Extended Data Fig. 8a**) and a lower lifetime exposure to tobacco smoking compared to the discovery dataset (P=0.041; **Extended Data Fig. 8b**). Despite these differences, the key findings from the discovery dataset were well replicated in the validation dataset. For instance, HAS tumors exhibited a significantly higher number of retrotransposon insertions, enriched *PIK3CA* and *TP53* driver mutations, higher expression of *APOBEC3A* and *APOBEC3B*, and higher expression of basal cell markers (*KRT15* and *TP63*) compared to LAS tumors (**Extended Data Fig. 8c-f**). LAS tumors were enriched with *STK11* and *KRAS* driver mutations and showed significantly higher expression of AT2 cell markers (*NKX2-1* and *SFTPB*) (**Extended Data Fig. 8d,f**). HAS tumors were dominated (80.4%) by the A3A-like mutator phenotype, whereas LAS tumors were dominated (51.6%) by the A3B-like phenotype (P=1.62e-23; **Extended Data Fig. 8g**). APOBEC3B was strongly associated with UNG expression in both LAS and HAS tumors (**Extended Data Fig. 8h**). The age at diagnosis was significantly higher in HAS tumors compared to LAS tumors (median 6.5 years difference; P=0.00086; **Extended Data Fig. 8i**). A positive correlation between APOBEC mutation ratio and age at diagnosis was also observed in the validation set (R=0.26; P=0.065; **Extended Data Fig. 8j**). We have now included these validation results in the last paragraph of our revised manuscript.

Unfortunately, the validation set lacked detailed tobacco smoking information, particularly for TTFC, preventing us from performing downstream association analyses between TTFC and mutational signatures, TMB, or DNA methylation of CpG probes. Among 184 subjects from the validation dataset, only 84 of them have information on the number of cigarettes smoked per day and the number of years of smoking. Thus, we are not able to replicate the findings related to other tobacco-smoking variables. We have de-emphasized our conclusions or removed these results from the manuscript. For example, we move the original **Figure 1b-c to supplementary figures**, as we could not replicate the association between Kategis and the APOBEC subtype. We highlighted these changes in the tracked version of our revised manuscript.

Subgroup Analyses and Statistical Corrections:

We understand the concern regarding multiple subgroup analyses and the associated risk of false positives. To address this, we applied the Benjamini-Hochberg procedure to control the false discovery rate (FDR) in our analyses where applicable (e.g., when multiple testing objects are independent variables). This method allowed us to manage the risk of type I errors while maintaining substantial statistical power. We have indicated FDR values or used FDR<0.05 as the threshold for significance in our figures (see **Fig. 1d, Fig.**

2a, Fig. 2c, Fig. 2d, Fig. 3c, Fig. 3f, Extended Data Fig. 2a, Extended Data Fig. 3, Extended Data Fig. 6, Extended Data Fig. 8d, Supplementary Fig. 6, Supplementary Fig. 16b). For tobacco smoking-related questions, we also applied multivariable analysis adjusting for additional covariates (see **Fig. 3b-f, Extended Data Fig. 5, Extended Data Fig. 6, Supplementary Fig. 10, Supplementary Fig. 11**). We applied a similar strategy to the new analyses in the validation dataset. We consulted multiple statisticians to ensure the appropriateness of our statistical methods and presentations. We have revised the manuscript to explicitly state and clarify the use of these correction methods in our analysis section.

Reviewer #2 (Remarks to the Author): expertise in lung cancer genomics

The study by Zhang et al performed multiomic profiling of 309 paired lung tumor-normal tissues, mostly lung adenocarcinomas (LUADs). By mutational signature analysis, the group found samples were divided mainly into two groups with low (LAS) and high (HAS) APOBEC mutagenesis. They found that LAS were enriched with A3B-like mutagenesis, KRAS mutations, higher mitotic rate, AT2/AT2-like expression and distal disease based on inference of cells-of-origin (COO), younger age of onset. In contrast, the HAS group were enriched with A3A-mutagenesis, TP53 mutations, lower mitotic rate, basal expression and proximal disease based on COO, and older age of onset. LAS had higher association with DNA repair (UNG, BER). LAS more likely showed associations between time to first cigarette (TTFC) and mutation burden as well as between time from quitting smoking and methylation (e.g., in AHRR) changes. HAS showed more predicted neoantigens than LAS. Given the above, the group construes that higher mutation burdens with lower DNA repair in HAS leads to senescence which then leads to mitotic arrest in presence TP53 mutations. This, along with the inferred lung regeneration occurring in HAS leads to a longer time to accumulation of mutations which, along with increased neoantigens that may indicate increased immune surveillance, explain the longer time to lung tumor onset. While the study represents a significant effort in multiomic profiling, this reviewer has major concerns with the speculative nature of the conclusions and some aspects in the study's design and presentation. The following comments need to be addressed prior to publication for the astute readership of Nature Communications.

Response:

We thank the reviewer for the detailed review and thoughtful comments on our manuscript. We appreciate the recognition of the effort involved in our multi-omic profiling study. We acknowledge the importance of clearly distinguishing between data-driven findings and hypotheses that require further validation.

As we responded to Reviewer #1's comments, we have added comprehensive validation analyses using an independent dataset (validation dataset, 184 newly sequenced TCGA LUAD tumor samples from smokers released in 2024). Compared to our original discovery dataset, we observed that the validation dataset had a significantly lower proportion of HAS tumors ($P=0.004$; **Extended Data Fig. 8a**) and a lower lifetime exposure to tobacco smoking ($P=0.041$; **Extended Data Fig. 8b**). Despite these differences in sample size, exposure, and APOBEC subtype proportion, the key findings from the discovery dataset were well replicated in the validation dataset as described in our revised manuscript (**Extended Data Fig. 8d-j**; see the response to Reviewer #1 and the last paragraph of the Results section).

We acknowledge that some original findings were not replicable in the validation dataset due to missing detailed tobacco information and the small sample size. In our revised manuscript, we have taken steps to ensure that conclusions that we could not test for validation are toned down and presented as hypotheses rather than definitive statements (see highlights in the tracked version of our revised manuscript). Additionally, we have revised the Discussion section to explicitly state which findings are supported by our data and which are speculative, providing additional context for the latter to highlight the need for further research. We believe this clarification will help readers understand the distinction and evaluate the implications appropriately.

-The study finds that LAS is more enriched with KRAS mutations, and HAS tumors are more enriched with TP53 mutations. While this appears to be the case (and the study postulates that these genomic differences are associated with differing genomic instability, cells of origin, among other variables), there are still quite a bit of TP53 mutations in LAS and KRAS variants in LAS. It is not clear then whether the paper suggests that KRAS mutant tumors in LAS and HAS are different (?). Another missed opportunity, perhaps concern here, is that TP53 and KRAS mutations typically co-occur (KP) and it has been shown that KP and KRAS mutant lung adenocarcinomas with wild type TP53 are almost completely different tumors. In this theme, is it not clear whether the biology of KP tumors is accounted for in the study's analysis and hypothetical conclusions.

Response:

We further investigated the relationship between driver genes and APOBEC subtypes as suggested by the reviewer. As shown in **Extended Data Fig. 1**, TP53 mutation frequencies were 68.1% in HAS tumors and 54.6% in LAS tumors. Among all TP53 driver mutations in our discovery dataset, only 28.7% (same mutation) were shared between HAS and LAS tumors, with the majority being exclusive to either HAS or

LAS. We have added bar graphs from the validation datasets as well as lollipop plots in **Extended Data Fig. 1** (and **Response Figure 1**) to highlight the differences in *TP53* mutation locations. We did the same for *KRAS* mutations.

As described in the manuscript, we found a significant enrichment of hotspot C>X mutations (X=any base) at TpC sites in *TP53* genes (22.8% in HAS, 14.2% in LAS, $P=1.37e-07$; OR=1.77). Similar results were observed when we limited the analysis to LUAD only. These results suggest that HAS tumors have a higher frequency and distinct pattern of *TP53* mutations compared to LAS tumors, potentially driven by APOBEC mutagenesis.

Overall, in our dataset, only 37 tumors had co-occurring *TP53* and *KRAS* mutations (12% overall; 9.6% in HAS and 13.8% in LAS). Similar frequencies were observed when limited to LUAD only (13.7% overall; 10.3% in HAS; 16.2% in LAS). Given the low frequency of tumors with co-mutations in *TP53* and *KRAS*, we do not believe this affects our conclusions. For validation, we removed these 37 tumors with co-occurring *TP53* and *KRAS* mutations from our data analysis. This exclusion did not change our major results (see **Response Figure 2**):

1. **APOBEC mutagenesis Enrichment in APOBEC Subtypes (Response Figure 2a):** Similar enrichment results were observed, with HAS tumors dominated by the A3A-like mutator phenotype and LAS tumors dominated by the A3B-like phenotype.
2. **Somatic Mutation Enrichment in APOBEC Subtypes (Response Figure 2b):** Similar enrichment results were observed, with *KRAS* enriched in LAS tumors and *TP53* enriched in HAS tumors.
3. **Association Between APOBEC3B and UNG Expression (Response Figure 2c):** We observed a significant association between *APOBEC3B* and *UNG* and no association for *APOBEC3A* and *UNG*, consistent with our manuscript's findings.
4. **Tobacco Smoking Variables and TMB (Response Figure 2d):** Time to first cigarette (TTFC) remained the only tobacco smoking variable significantly associated with TMB, as described in the manuscript.
5. **Expression of Epithelial Markers and APOBEC Subtypes (Response Figure 2e):** LAS tumors showed higher expression of AT2 cell markers (*NKX2-1*, *NAPSA*, and *SFTPB*), while HAS tumors exhibited higher expression of basal cell markers (*KRT19* and *TP63*).
6. **Age at Diagnosis (Response Figure 2f-g):** Significant differences in age at diagnosis were observed between HAS and LAS tumors. HAS tumors had a significantly later age at diagnosis (median 3.2 years difference). Among heavy smokers (TTFC<5 mins), the age at diagnosis difference was approximately 8.6 years.

Thus, we believe that the tumors with co-occurring *TP53* and *KRAS* mutations do not affect our conclusions. We have now added this point to our Discussion in the revised manuscript.

-There are various differences in measures between groups accentuated by the study that are at best modest, such as the association between short TTFC and mutated genes associated with smoking (ED Fig. 7), expression of *KRT15*, *TP63* between LAS and HAS (Fig. 4a), etc.

Response:

We understand the concern regarding the modest differences observed between groups for certain measures. However, we want to emphasize the strong consistency of findings across orthogonal approaches and multiple data types. Our original discovery dataset consists of 309 tumor samples with full tobacco smoking information available for 195 tumors, including five independent smoking variables (to our knowledge there is no additional dataset with WGS data from lung cancer and detailed tobacco smoking information, e.g., including TTFC). We applied rigorous statistical approaches for all analyses. For example, we used linear or logistic regression analysis between LAS and HAS tumors adjusting for multiple covariates, including age, sex, histology, TMB, and tumor purity (**Fig. 3**). Although the statistical power was limited, the observed differences were statistically significant and consistent with our overall hypothesis. The reviewer cites the association of TTFC and mutated genes. This association with TTFC was significant not only in association with driver gene mutations but also in association with additional genomic features in accord with our hypothesis. For example:

- TMB (**Fig. 3b, Supplementary Fig. 10**)
- SBS4 (**Extended Data Fig. 5a**)
- DBS2 (**Extended Data Fig. 5b**)
- ID3 (**Extended Data Fig. 5c**)
- Different mutation types (**Supplementary Fig. 11**)
- *ZFHX4* mutation status (**Fig. 3c-d**)

All these associations were observed only in LAS tumors, not in HAS tumors. Additionally, **Fig. 3e-f** highlights the association between time from quitting smoking (CIGT_TIME_LAST_QUIT) and CpG methylation levels of the *AHRR* gene observed only in LAS tumors, again supporting our hypothesis.

Similarly, consistent results were identified using gene expression analyses. Our discovery dataset includes 309 subjects, with 183 having both WGS data and RNA-Seq expression data. We selected multiple well-known markers (*NKX2-1*, *NAPSA*, *SFTPB*, *KRT19*, *KRT15*, *TP63*) to test the difference between cell types

to ensure the consistency of our findings. We also performed pathway analysis as shown in **Supplementary Fig. 16**, which again highlights the cell type differences between HAS and LAS. Importantly, the enrichment of LAS tumors with AT2-cell like types and HAS tumors with basal cell-like types was based not only on RNA-Seq expression data, but also on methylation data (**Fig. 4b**) and WGS data (COO analysis, **Supplementary Fig. 18**).

The consistent findings from different data types and analytical approaches point to real underlying biological processes and contribute to our understanding of the distinct characteristics of LAS and HAS tumors.

-On that theme, this reviewer has major concerns with the COO analysis which is based on a method by the group currently under peer review and there are insufficient details in the Methods with regards to this analysis. The study suggests that Club, basal, ciliated cells are enriched in HAS whereas distal AT2 cell types are enriched in LAS. Review of z-scores in Supp. Fig 18 suggests that the pulmonary healing signaling pathway (how many genes are in this pathway? and what are the genes?) is activated in both LAS and HAS. It is also difficult to make sense of Supp. Fig. 19 since most tumors in LAS and HAS show an ambiguous COO and the associations in the heatmap in Supp. Fig. 19a (not sure if the association scale is correct) are not clear. The discussion section mentions that LAS is enriched with KRAS mutations and that AT2 cells are likely the cells of origin of KRAS mutant tumors. While this may be true, many reports have shown that KRAS mutant LUADs have reduced alveolar differentiation and in fact lose AT2 markers (e.g., mucinous KRAS-mutant LUADs). Additionally, various studies have shown that KRAS mutations in AT2 cells are drivers of regeneration in alveolar niches. These previously published reports are contradictory to the conclusions (e.g. LAS low regeneration more KRAS, HAS high regeneration less KRAS) of the study. Since the study centers on cohorts that are richly annotated, could the group review the clinicopathological data to determine anatomical (distal versus proximal) of LUADs in the LAS and HAS groups?

Response:

We thank the reviewers for these comments. We have addressed them and made significant updates to the manuscript.

Firstly, We have revised the Methods section to include detailed information on the cell-of-origin (COO) analysis and have re-run the analysis using updated COO algorithms. This has significantly improved our ability to assign cell types to each tumor, as shown in the new **Supplementary Fig. 18**. We provide a comprehensive explanation of the association and the scale in the updated figure legend and methods section.

In **Supplementary Fig. 16**, we performed multivariate analyses to identify genes whose expression was associated with TTFC after adjusting for tumor purity and copy number status. Regression analyses were performed separately within the LAS and HAS subtypes. Genes with an association $P < 0.05$ were selected for pathway analyses using Ingenuity Pathway Analysis (IPA). The most significantly upregulated pathway associated with short TTFC in HAS tumors was the pulmonary healing signaling pathway. Although this pathway was also found to be associated with short TTFC in LAS tumors, the effect size was much smaller (Z -score HAS=3.78 vs. LAS=1.89). There were 28 genes involved in the pulmonary healing signaling pathway with expression significantly associated with TTFC (**Response Figure 3**), including Vascular endothelial growth factor A (*EGFA*), epidermal growth factor receptor (*EGFR*), Matrix metalloproteinase (*MMP1*), mitogen-activated protein kinase (*MAPK10*), Wnt family members (*WNT5A*), SMAD family member (*SMAD5*), MYC proto-oncogene (*MYC*), Catenin beta 1 (*CTNNB1*), elastase neutrophil expressed (*ELANE*), BLK proto-oncogene Src family tyrosine kinase (*BLK*), heparin-binding EGF like growth factor (*HBEGF*), etc. We added a direct comparison of the expression levels of these genes in the pulmonary healing signaling pathway and found that the pathway was up-regulated/activated in HAS tumors (Z -score=1.57; $P=1.06e-03$) compared to LAS tumors. Specifically, 19 of the 28 genes were differentially expressed between HAS and LAS ($FDR < 0.05$; **Supplementary Fig. 16b**). This supplementary figure has been added to the manuscript.

With consistent results across multiple data analyses (COO analysis from WGS data, marker gene expression, and pathway analysis from RNA-Seq), our findings suggest that proximal lung cell types (Club/Basal/Ciliated cells) are enriched in HAS, while distal lung cell types (AT2) are enriched in LAS.

As mentioned by the reviewer, we recognize that it is challenging to identify the cell of origin of LUAD in humans. While there are reports showing that *KRAS* mutant LUADs have reduced alveolar differentiation, numerous studies discuss *KRAS* mutations originating from alveolar type II (AT2) cells. For example:

1. Chaudhary et al. (2023) highlight that activation of *KRAS* G12D in AT2 cells leads to cellular proliferation and progression into lung adenocarcinoma, with these cells exhibiting high plasticity and tumor-initiating potential. (PMID: 37882674)
2. Dost et al. (2020) demonstrate that AT2 cells with oncogenic *KRAS* show reduced expression of mature lineage genes, emphasizing the early transcriptional changes induced by *KRAS* activation in AT2 cells. (PMID: 32891189)
3. Desai et al. (2014) highlight the role of AT2 cells as stem cells contributing to alveolar renewal and repair, which transform into cancer cells upon *KRAS* activation. (PMID: 24499815)
4. Naranjo et al. (2021) developed murine AT2 cell organoids resembling human LUAD, particularly those driven by *KRAS* mutations, maintaining AT2 cellular identity. (PMID: 36175034)

When we compared the expression levels of AT2 cell markers using RNA-Seq data by *KRAS* mutation status in our study, we consistently observed higher expression of AT2 cell markers in *KRAS* mutant tumors, regardless of the APOBEC subtype (LAS/HAS) (**Supplementary Fig. 17a**). Similar results were also observed in COO analysis comparing *KRAS* mutant and wild-type tumors (**Supplementary Fig. 17b**). These data strongly link *KRAS* mutations with AT2 cells in LUAD. However, there are additional reports showing that in certain conditions, expression of *KRAS*(G12D) in differentiated AT1 cells reprograms them slowly and asynchronously back into AT2 stem cells (PMID: 37468622). So, it appears that *KRAS*

mutations can induce AT2 cells directly (with a resulting aggressive behavior) or indirectly through AT1 reprogramming into AT2 cells (with indolent behavior). Given the complexity, we have added the following sentence to the Discussion:

“Although our study supports an AT2-like cell of origin of KRAS mutant tumors, further investigation on the link between AT2 cell types and KRAS mutations is warranted.”

Per the reviewer’s request, we reviewed the clinicopathological data. Unfortunately, we do not have data on tumors’ locations with respect to proximity to the bronchial area vs. distal area. We only have data on whether the tumors were on the upper or lower lobes of the lung. As expected, we found no significant difference between HAS and LAS in terms of upper or lower lobes ($P=0.15$; **Response Figure. 4**).

-The study construes that HAS tumors, given the higher neoantigens, could less likely evade host immunity which could partially explain the longer time (higher age of patients) lung tumor onset. This is a conclusion that while being interesting is also speculative. Could the group deconvolute the RNA-seq data to determine whether HAS tumors have higher immune infiltration compared with LAS tumors?

Response:

Following the reviewer’s suggestion, we applied RNA-seq data to assess immune cell infiltration. Among our tumor samples with both RNA-Seq data and WGS data, there are 51 LAS and 41 HAS tumors. We utilized the Xcell tool to dissect the cellular heterogeneity within the tumor microenvironment and compare the immune infiltration between HAS and LAS. Our analysis focused on immune cells with an overall proportion greater than 0.1. Despite the power limitations, our analysis revealed that the HAS group indeed exhibits a higher proportion of T cell CD4+ Th1 ($P=0.048$) and T cell CD4+ Th2 ($P=0.087$) cells, as well as common lymphoid progenitors ($P=0.1$). Conversely, the LAS group demonstrated elevated levels of hematopoietic stem cells and a higher stroma score ($P=0.1$). These findings suggest that the APOBEC signature is associated with a more robust immune response in HAS tumors, as evidenced by the presence of immune cells known for their anti-tumor activity. This supports our hypothesis that higher neoantigen levels in HAS tumors might lead to increased immune surveillance, potentially contributing to the longer time to lung tumor onset observed in these patients. We have included this analysis in **Extended Data Fig. 7**.

-PD-L1 RNA is modestly different between LAS and HAS. Could PD-L1 expression differences be confirmed in a subset of LAS and HAS cases using standard (and more appropriate for PD-L1) immunohistochemistry?

Response:

Unfortunately, we do not have immunohistochemistry data. However, we examined the lymphocytic infiltration levels using the available H&E slides from 38 tumors (19 HAS and 19 LAS). We scored the

level of lymphocytic infiltration on a scale of 1-5 and classified the tumors as having high or low lymphocytic infiltration based on a threshold score of 2. As expected, we found that HAS tumors had a higher proportion of high lymphocytic infiltration than LAS tumors (HAS 42.1% vs. LAS 26.3%; see **Extended Data Fig. 7**), although the difference was not statistically significant.

-The presentation of the study is not adequate, there are very few panels/data in the main figures (are these the only main talking points?) and the data in the ED and supp figures are too sparse and could easily be consolidated.

Response:

We appreciate the reviewer's feedback on the presentation of our figures. We understand that the clarity and organization of figures are crucial for communicating research findings effectively. To address the concerns raised, we have reorganized the layout and content of our main and extended data figures. We hope these changes will make the data presentation more efficient and the narrative of our findings clearer to the readers.

Reviewer #3 (Remarks to the Author): expert in cancer bioinformatics and evolution analysis

This study presents a multi-omics characterization including WGS of hundreds of lung tumors newly processed and data generated. Patterns of APOBEC mutagenesis are analyzed in the context of smoking phenotypes and mutational signatures, and in the context of gene expression patterns and DNA methylation patterns.

Regarding the association of high-APOBEC or low-APOBEC mutagenesis tumors with mutations in certain driver genes (here, TP53 and KRAS) – associations of signatures with driver events were addressed in various prior studies e.g. PMID: 30412573, PMID: 29748584. Implications of these TP53 and KRAS associations with APOBEC mutagenesis to cancer evolution are not clear. Similarly, concluding that APOBEC may have caused some of these mutations just based on the context enrichment is perhaps premature.

Response:

We thank the reviewer for the insightful comments. We appreciate the recognition of the effort and the sample size of our multi-omics characterization of lung tumors. We address the specific concerns below:

Other studies, such as those referenced by the reviewer (PMID: 30412573, PMID: 29748584), have investigated the association between mutational signatures and driver events. However, these studies have not specifically explored such associations in the context of a strong exposure to tobacco smoking. As highlighted by these papers, different mutational processes can be associated with the same driver mutations.

Our study introduces a novel perspective by investigating the association between APOBEC mutagenesis and driver gene mutations in the context of tobacco-smoking signatures. Specifically, in our study of lung cancer in smokers (with SBS4 signatures), we found that *KRAS* mutations are enriched in LAS tumors, while *TP53* mutations are enriched in HAS tumors. Indeed, we also found a strong association between the higher APOBEC group (HAS) and *PIK3CA* mutations, which has also been reported in both of the referenced papers (PMID: 30412573, PMID: 29748584). Although *KRAS* mutations have been previously associated with the SBS4 signature, our results suggest that this association may be influenced by the APOBEC mutagenesis. Indeed, we found that the interplay between the APOBEC and tobacco smoking mutagenesis has significant implications for tumor evolution (slow growth, late age at onset of the tumor), as described in our manuscript using multi-omics data analyses.

We do not emphasize the role of driver genes in tumor evolution. Indeed, as shown in **Supplementary Fig. 20**, we found no significant difference in the age at diagnosis between tumors with *KRAS* or *TP53* mutations. We believe it is the interaction between the APOBEC mutagenesis and tobacco smoking mutagenesis that has a strong indirect influence on tumor evolution. The enrichments of *TP53* in HAS and *KRAS* in LAS are primarily driven by different combinations of mutagenesis processes.

Similarly, our manuscript does not claim that APOBEC directly causes mutations in these driver genes. We simply report the association between the APOBEC groups (HAS vs LAS) and the driver gene mutations, although we do highlight the enrichment of C>X mutations at TpC sites in the driver genes (*TP53/PIK3CA*), a genomic context similar to that of the APOBEC mutational signatures. Further studies or experimental validations are needed to establish a causal relationship between the APOBEC activity and these driver gene mutations.

Regarding the association of UNG expression levels with APOBEC3B (but not 3A) expression; it was not explored how this association bears upon APOBEC mutagenesis levels – does the UNG actually prevent mutagenesis? The statement „This finding suggests that UNG expression is activated by uracils in DNA generated by APOBEC3B mutagenesis or that UNG and APOBEC3B share common regulatory

mechanisms“ is highly speculative and joins the plethora of other heavy speculation sprinkled through the Results section. The coexpression analysis gives no indication of direction of effect or confounding. Attempting to control for subtype and impurities would help make this result firmer. Also, it is not clear that the association of $P=8e-4$ would stand after a FDR correction – I presume that many genes were tested.

Response:

As many studies have shown, the A3B-catalyzed uracil lesions are repaired by uracil DNA glycosylase (UNG)-driven base excision repair. Consistent with these findings, our analysis identified a strong positive correlation between the expression of *APOBEC3B* and *UNG* in both LAS and HAS tumors. However, we acknowledge that the expression data reflect the status at the time of diagnosis, while the APOBEC mutagenesis can be episodic.

Following the reviewer’s comment, we conducted additional analyses to explore the effect of *UNG* expression on APOBEC mutagenesis (**Response Figure 5**). First, we evaluated the relationship between *UNG* expression and the estimated load of APOBEC mutations using the P-MACD method. We found no significant associations between *UNG* expression and APOBEC mutagenesis in both HAS and LAS tumors. Next, we performed a statistical regression analysis in both LAS and HAS tumors using the following formula:

APOBEC mutagenesis level \sim *UNG* expression + *A3A* expression + *A3B* expression + covariates (tumor purity, histology, sex, age)

We used the estimated load of APOBEC mutations as the APOBEC mutagenesis level. The results showed that in both LAS and HAS tumors, *UNG* was negatively associated with APOBEC mutagenesis levels, but the association was not significant. Thus, from our dataset, we cannot conclude that *UNG* prevents APOBEC mutagenesis. Further investigation is needed to evaluate this relationship. Following the reviewer’s comment, we have deleted the following sentence: “This finding suggests that *UNG* expression is activated by uracils in DNA generated by APOBEC3B mutagenesis or that *UNG* and APOBEC3B share common regulatory mechanisms.”

Regarding the co-expression analysis, we did not perform FDR correction based on all human genes. According to the literature, there is substantial evidence that APOBEC-catalyzed uracil lesions are repaired by UNG-driven base excision repair (PMID: 10767624, PMID: 24746924). Our hypothesis is that UNG repairs the mutations generated by *APOBEC3B* or *APOBEC3A*, but not other APOBEC enzymes. Therefore, we applied FDR correction only to the APOBEC genes (N=10). After FDR correction, the correlation between *APOBEC3B* and *UNG* remained highly significant in both HAS and LAS tumors (LAS: R=0.35, P=4.75e-04, and FDR=5.23e-03; HAS: R=0.46, P=8.52e-06, and FDR=9.37e-05). We have updated this information by adding FDR values in the Results section.

I find the logic in the clonality commentary is not well explained; it is not clear how clonality supports episodic activity of APOBEC or lack thereof, nor is episodic activity itself sufficient reason for lack of correlation to gene expression.

Response:

We thank the reviewer for this comment. We understand that the logic behind the clonality commentary may not have been clearly explained. Here, we aim to clarify our reasoning:

According to the literature, APOBEC mutagenesis can occur in bursts of rapid mutation accumulation ('episodic mutagenesis') followed by longer periods with little mutation activity, as shown, for example, in cell lines (e.g., PMID: 30849372). In tumor studies (e.g., PMID: 30840888), APOBEC signatures vary within each patient, sometimes being absent in primary tumors but highly prominent in some metastases. These studies support the idea that episodic APOBEC mutagenesis may occur in the late stages of tumor development (subclonal mutations).

In our manuscript, we stated, "The clonality of APOBEC mutations does not support frequent episodic APOBEC mutagenesis as the likely explanation for the lack of association between *APOBEC3A* and *UNG*/BER expression." As shown in **Extended Data Fig. 4**, we found that most tumor samples were dominated by clonal APOBEC mutations, both in our study and the PCAWG study. If tumors experienced very frequent episodic APOBEC activity bursts, we would expect to see APOBEC mutations accumulating at different times, including during subclone expansion stages. The fact that many samples (HAS tumors) exhibit only clonal mutations and not subclonal mutations, does not support the hypothesis of frequent episodic APOBEC activity.

RNA expression data capture the gene expression landscape at the time of diagnosis. During this snapshot, APOBEC mutagenesis may be very quiet (resulting in very few mutations), which could not trigger the base excision DNA repair process and subsequent induction of *UNG*. This could be a reason for the lack of correlation between *APOBEC3A* and *UNG*. However, we acknowledge that other factors or regulatory mechanisms could also be influencing the observed patterns.

Regarding „as well as a higher proportion of APOBEC mutations contributing to kataegis (61.4% versus 50.8%; Fig. 1d-e), even in clusters of very low numbers of mutations (41.7%).“ Are the clusters with low numbers of mutations omikli? They would need to mention them as such (they do so in the methods).

Response:

The phrase "clusters with low numbers of mutations" refers to kataegis clusters with very few mutations (kataegis clusters < 4 mutations). In HAS tumors, 41.7% of APOBEC mutations contributed to these small clusters with fewer than four mutations. Conversely, the remaining APOBEC mutations (61.4% - 41.7%) contributed to clusters with a larger number of mutations (≥ 4). Additionally, since we could not replicate the enrichment of kataegis in HAS tumors compared to LAS tumors in the validation dataset (new TCGA

LUAD WGS dataset), we have removed the sentence of “even in clusters of very low numbers of mutations (41.7%).” to avoid any confusion and also moved all relevant figures to the supplementary figures.

The section „Tobacco smoking addiction is associated with genomic changes only in LAS tumors“ which appears central to the story (suggesting interaction between smoking and APOBEC signatures) has some premature conclusions and speculation. For example, why would the “time to first cigarette (TTFC) in the morning, a marker of strong nicotine addiction“ be a better measure than more direct measures of tobacco exposure seems odd (their explanation of better DNA repair in the morning seems a bit of a stretch). Next, in the high-APOBEC tumors they do not find an association between smoking exposure and smoking signature, which they explain by a non-parsimonius mechanism (lines 208-212), involving a lot of steps for which there appears not to be supporting data. Some of the steps do not even appear plausible at an initial inspection, such as TP53 mutant cells having more apoptosis (mutation in TP53 should reduce ability to apoptose not vice versa). The general idea is that APOBEC3A activity kills cells, and for some reason preferentially clearing the tobacco smoke mutant ones. Unfortunately this is not the most convincing in the absence of more specific data supporting it.

Response:

We thank the reviewer for this comment, which gives us the opportunity to clarify and expand on our findings.

We found that TTFC was the strongest mutagenic measure of tobacco smoking exposure in our study. Cigarette smoking delivers nicotine to the brain within seconds, facilitating nicotine dependency. Addiction to tobacco smoking is widely assessed through the Fagerstrom Test for Nicotine Dependence. Among the six common tobacco smoking variables, we and others have previously identified TTFC as a strong risk factor for lung cancer independent of other measures of tobacco smoking exposure (PMID: 24948709). TTFC has been considered a single-item measure of nicotine dependence, biological uptake of nicotine, and smoking cessation success. It also has a dose-dependent relationship with the tobacco-specific carcinogen urinary NNAL (4-methylnitrosamino-1-3-pyridyl-1-butanol) (PMID: 23542804).

Mechanistically, the effects of shorter TTFC are possibly attributed to “compensatory smoking,” a tendency to inhale cigarette smoke more deeply in the morning to quickly reach the needed nicotine levels after overnight smoking cessation. The sudden, intense impact of carcinogens on the lung epithelia can result in high DNA damage that could challenge the DNA repair capacity, which is lower in the morning (PMID: 25728089). We hypothesize that the stronger effect of TTFC on mutagenesis could be related to a deeper inhalation and a lower capacity to repair the DNA damage due to the circadian rhythm in the morning, more than a tobacco dosage effect. Moreover, circadian disruption has previously been implicated in lung cancer tumorigenesis. Both catecholamine and glucocorticoid secretions are regulated by the circadian rhythm, and evidence suggests that their release is accelerated by components of cigarette smoke. When nicotine binds to nicotinic acetylcholine receptors expressed on adrenal medulla cells, it triggers the release of

catecholamines into the bloodstream, which may promote epithelial cell proliferation, contributing to cell malignant transformation. Further investigations with large-scale sequencing studies and detailed smoking information, including TTFC and inhalation habits, are required to confirm our hypothesis.

In our manuscript, every analysis we have conducted follows logical steps to verify our hypothesis and explain the results. We used multiple approaches and data types to prove our hypothesis. This study is based on correlation but the large number of consistent results across approaches substantiate our claim. Our proposed model, shown in **Figure 5**, aims to explain the observations in HAS tumors through a comprehensive cellular dynamic process. This model is supported by various multi-omics data analyses. All major results have been validated in the newly released TCGA LUAD WGS dataset, along with updated methods for cell-of-origin analysis.

We do not emphasize the role of TP53 in reducing or activating apoptosis directly in our model, although several papers do show that p53 mutants retain partial function and can induce apoptosis under certain conditions (PMID: 16543937; PMID: 9841917; PMID: 11790556; PMID: 11058599). Instead, we suggest that hypermutation by A3A and tobacco smoking mutagenesis, combined with limited DNA repair capacity and TP53-induced genomic instability, can trigger senescence, apoptosis, and cell regeneration. This is indicated by high expression of the pulmonary healing signaling pathway, stemness markers, and distal cell-of-origin in HAS tumors. Mechanistically, single-cell transcriptomics analyses in mice models have shown that during LUAD evolution, p53 promotes AT1 differentiation through action in a *transitional cell state*. Notably, p53 also directs *alveolar regeneration* after injury by regulating AT2 cell self-renewal and promoting transitional cell differentiation into AT1 cells. Thus, p53 governs alveolar differentiation (PMID: 37468633) and this can contribute to the higher stem-like phenotype of the HAS (TP53-enriched) tumors. We have added this explanation and related references in the revised manuscript to further clarify our conclusions.

The DNA methylation analysis in the lines 213 – 237 is interpreted such that lack of association between certain probes and smoking phenotypes in the high-APOBEC tumors somehow suggests changes in cell turnover, however the lack of association could result from a various technical and biological sources of noise.

Response:

We appreciate the reviewer's interest in our findings regarding methylation markers and their association with smoking phenotypes. The identification of these associations exclusively in lung cancer patients with high APOBEC activity corroborates the evidence from whole-genome sequencing and transcriptomic data. To address potential technical variability, we have rigorously applied quality control measures across all omics data analyses. Specifically, in the methylation dataset, we employed rigorous steps for batch corrections and normalization (see Method section). We also verified that there was no significant difference across residence or occupational exposures across these individuals, which should minimize potential environmental effects explaining these differences. Furthermore, we conducted multivariable

analyses including different smoking variables as well as major potential confounders (age, sex, tumor histology, and tumor purity). Finally, the most significant probe (AHRR cg05575921) we identified is the very well-known CpG site associated with smoking cessation (PMID: 28100713). Thus, we concluded that these findings reflect true biological differences, such as increased cell turnover, rather than artifacts.

That APOBEC (or whatever mutagen) generates more neoantigen is somewhat unsurprising given that APOBEC mutagenesis is known to be enriched in early-replicating, genic regions, compared to many other mutagens (esp. tobacco smoke) where mutations are depleted in early-replicating DNA.

Response: We agree with the reviewer's comment and acknowledge that this has been reported by other studies (e.g., PMID:26527001). APOBEC-induced mutations occur preferentially in early-replicating, gene-dense, and active chromatin regions, likely due to the higher propensity of these regions to form single-strand DNA substrates for APOBEC enzymes. This characteristic could explain the high propensity for generating neoantigens. The increased neoantigen load in HAS tumors likely enhances tumor immune editing during development, potentially contributing to their later age at diagnosis.

Association of high-APOBEC mutagenesis with higher age is potentially interesting – did this replicate in the TCGA (or another cohort)? About their claim „These findings suggest that APOBEC mutagenesis has a stronger effect on the tumor clonal expansion [...] than on the subsequent tumor progression.“ it is not clear how this claim of clonality connects to the age-analysis section; maybe it is a reference to the VAF analysis (lines 180-189).

Response:

Yes, we were able to validate our findings using the newly sequenced TCGA LUAD WGS data. As added to our last Results section, “Validation in TCGA LUAD WGS Data released in spring 2024” we successfully replicated the association between high-APOBEC mutagenesis and higher age at tumor onset in this new dataset, with a significant p-value of 0.00086 and median 6.5 years difference between LAS and HAS tumors.

Regarding the claim, “These findings suggest that APOBEC mutagenesis has a stronger effect on the tumor clonal expansion [...] than on the subsequent tumor progression,” we aimed to explore which stage of tumor development is more strongly affected by APOBEC mutagenesis. To do this, we estimated the age at the appearance of the most recent common ancestor (MRCA), which reflects the clonal expansion, and also calculated the latency (difference between age at diagnosis and age at MRCA), reflecting the subclonal diversification or tumor progression.

As shown in **Supplementary Figure 19**, the MRCA exhibited a stronger difference between HAS and LAS (overall and in the group with short TTFC) compared to latency. These findings suggest that the APOBEC mutagenesis has a stronger effect on tumor clonal expansion, rather than on the subsequent tumor progression.

Typo „inclineuded“.

Response:

We are grateful to the reviewer for identifying this typographical error. It has been corrected in the revised manuscript.

Responses to reviewers

Reviewer #1 (Remarks to the Author):

I have no additional comment. The authors have responded adequately to my queries and suggestions.

Reviewer #3 (Remarks to the Author):

In the revised study, the authors have included a replication cohort, and reported several additional findings. There is some novelty in association of high-APOBEC tumors ("HAS") with age, however for some other findings reported, they instead confirm existing work. For instance, regarding link between A3 mutagenesis and immune infiltration added in revision: this mirrors prior reports of links between APOBEC expression and/or mutagenesis with tumor immunogenicity as reported for lung cancer and other cancer types (e.g. PMID:29695832, PMID: 31222843, PMID: 37215984).

Response:

We thank the reviewer for highlighting the connection between APOBEC mutagenesis and tumor immunogenicity, as reported in prior studies. Specifically, PMID: 29695832 emphasizes the significant association between APOBEC3B upregulation and immune gene expression, as well as biomarkers for immunotherapy response. Similarly, PMID: 31222843 discusses the relationship between APOBEC mutational signatures and immune checkpoint blockade (ICB) therapy response, while PMID: 37215984 explores similar effects of APOBEC mutagenesis on immune infiltration and immunotherapy response in esophageal squamous cell carcinoma (ESCC).

While these studies underscore the importance of APOBEC-driven mutagenesis in tumor immunogenicity, our manuscript uniquely compares the role of APOBEC3A- vs. APOBEC3B-driven mutagenesis in lung cancer. Specifically, we demonstrate that tumors characterized by an A3A-like mutational pattern exhibit stronger associations with tumor immunogenicity compared to A3B-like tumors. Our study focuses on lung cancer in smokers, linking APOBEC mutagenesis to tobacco exposure. Tobacco-related carcinogens may interact with APOBEC-driven mutagenesis, amplifying neoantigen generation in HAS compared to LAS, making HAS tumors potentially more responsive to immune checkpoint blockade therapies compared to LAS tumors. In addition, the heightened immune response may contribute to the slow growth of HAS tumors. These results add a new layer to the understanding on how APOBEC-related processes, potentially modulated by smoking, shape tumor immunogenicity. We have now cited

these references in our manuscript to place our findings within the context of prior work and highlighted the unique contribution of our study.

Next, links between A3 mutagenesis and mutations in *PIK3CA* and *TP53* drivers were reported here as well as previously (*PIK3CA* in PMID: 35013316, PMID: 30412573; *TP53* in breast and head&neck cancer in PMID: 37922356).

Response: We thank the reviewer for pointing out the links between A3 mutagenesis and mutations in *PIK3CA* and *TP53* drivers, as reported in prior studies of different cancer types. We fully agree with this observation and have now included these references in our manuscript to further contextualize and reinforce our findings on the enrichment of *PIK3CA* and *TP53* mutations in HAS tumors.

Further findings related with association of high-APOBEC tumor genomes (here "HAS") with A3A and/or A3B gene expression levels are indeed numerous in the prior literature. The reported positive association of A3A-versus-A3B signature balance with overall A3 mutation burden ("HAS tumors were dominated (80.4%) by the A3A-like mutator phenotype, whereas LAS tumors were dominated (51.6%) by the A3B-like phenotype") mirrors the previously reported findings (PMID: 26258849). An important issue for authors to seriously consider is presenting various findings as novel either explicitly or having context suggesting that, while they were reported before.

Response:

Thanks for highlighting the prior literature on the association of high-APOBEC tumor genomes (HAS) with A3A and/or A3B gene expression levels, as well as the positive association of the A3A-versus-A3B signature balance with overall APOBEC mutation burden. We acknowledge that our use of the same approach as described in PMID: 26258849 to identify A3A-like and A3B-like tumors is consistent with previously reported findings. This method was instrumental in understanding the differences in mutator deaminase activity between HAS and LAS tumors, as highlighted in our original manuscript.

While we recognize that the link between the genome of APOBEC-high tumors and the expression of *APOBEC3A* and/or *APOBEC3B* is not entirely novel, these results are critical in the context of our study. Specifically, they provide a foundation for elucidating the differences between high APOBEC mutagenesis (HAS) and low APOBEC mutagenesis (LAS) tumors in the context of exposure to tobacco smoking, particularly in terms of mutagenesis activity, driver gene mutations, and immune infiltration. To address

the reviewer's concern, we have cited the relevant references and revised our manuscript to appropriately clarify prior knowledge and new contributions, ensuring that context and significance are accurately conveyed.

Further, there remain issues to be sorted out regarding statistical support of findings, which presents, essentially, anecdotal examples in place of rigorous, systematic tests. This is of course convenient for getting passable p-values/FDRs however they should consider a more rigorous approach. Examples thereof would be as follows: (i) "HAS tumors exhibited... higher expression of basal cell markers (KRT15 and TP63)." and "LAS tumors... showed significantly higher expression of AT2 cell markers (NKX2-1 and SFTPB)." Which other genes have a stronger association than these hand-picked markers?

Response:

We appreciate the reviewer's insightful comments regarding the statistical rigor of our findings related to Figure 4a.

Initially, the LUAD cell type-specific markers referenced in our study were derived from a recent single-cell RNA-Seq analysis of LUAD (PMID: 34764257). To enhance the rigor of our approach, we have performed differential expression analysis between LAS and HAS tumors using all detected protein-coding genes from our bulk RNA-Seq dataset. Additionally, we compiled a comprehensive set of cell markers for human AT2 and basal cells from the literature included in the CellMark2.0 database (PMID: 36300619).

Our analysis revealed that among 18 AT2 markers and 95 basal cell markers with available expression data, 10 AT2 markers (55.6%) were consistently upregulated in LAS tumors, while 72 basal cell markers (75.8%) were upregulated in HAS tumors. Compared to other genes, many AT2 and basal cell markers displayed either highly significant p-values or large fold-change values in our volcano plots. These results further corroborate our original findings.

To address the reviewer's question regarding other genes with stronger associations than these markers, we conducted an IPA pathway analysis on the genes most strongly associated with APOBEC subtypes. This analysis highlighted pathways related to most significant molecular and cellular functions: "Cell Death and Survival," "Cellular Function and Maintenance," and "Cellular Development," which align with the expected upregulation of these pathways in HAS tumors as described in our manuscript. For example, among the top significant genes, **PDCD10 (Programmed Cell Death 10, P = 1.73e-05)**: This gene is directly involved in programmed cell death and cell survival signaling pathways. **PRDX1 (Peroxiredoxin 1, P = 5.4e-06)**: Plays a role in protecting

cells from oxidative stress-induced apoptosis. **IL18 (Interleukin 18, P = 1.84e-05):** Associated with inflammation and capable of modulating cell death pathways.

We have now included these analyses and the corresponding volcano plots in the revised manuscript (**Supplementary Fig. 16c**). These additions provide a more comprehensive and statistically robust foundation for our findings and clarify the associations highlighted in our study.

(ii) "We observed a significant association between APOBEC3B and UNG and no association for APOBEC3A and UNG" How many other DNA repair or related genes have a stronger association with A3B mutagenesis (or A3A/B expression) than UNG does? The choice of UNG is based on some prior knowledge but many other genes could have overlapping mechanisms in the BER or MMR pathways.

Response: In **Extended Data Fig. 3** of our original manuscript, we had analyzed 32 essential genes in the base excision repair (BER) pathway, as identified from the literature. Among these, 21 genes showed a significant association with *APOBEC3B* expression after multiple testing correction (FDR < 0.05 based on number of BER genes). Notably, UNG emerged as the most significantly associated gene with *APOBEC3B* expression, regardless of the APOBEC subtype, with a multiple testing FDR of 2.5e-08. Following the reviewer's question, we additionally examined genes in the mismatch repair (MMR) pathway (e.g., *MSH2*, *MSH3*, *MSH6*, *MLH1*, *MLH3*, *PMS1*, *PMS2*), and none of the MMR genes exhibited a stronger association with *APOBEC3B* expression than *UNG*. This highlights the robust relationship between *UNG* and *APOBEC3B*.

(ii-b) Related with this, another concern is that UNG association is FDR-corrected only over the APOBEC paralogs (n=10) however this may mislead because the association test is not to distinguish one APOBEC paralog from another but rather one DNA repair gene from another, and DNA repair genes are >>10.

Response: We appreciate the reviewer's insightful comment. In fact, in **Extended Data Fig. 3**, the FDR correction was already performed based on the total number of BER genes. To further clarify this point, we have now revised **Fig. 2c** by performing multiple testing correction across all tested BER genes (N=32). In the updated figure, the association between *APOBEC3B* expression and *UNG* expression remains statistically significant in both LAS (**FDR = 0.014**) and HAS tumors (**FDR = 2.72e-04**).

We have updated the manuscript and figure legend accordingly, ensuring greater clarity and alignment with the reviewer's concern.

(iii) There is implication that statistical interactions between genetics and smoking variables have been tested when this seems not to be the case "Other studies [...] have not specifically explored such associations in the context of a strong exposure to tobacco smoking.... Our study [...] investigating the association between APOBEC mutagenesis and driver gene mutations in the context of tobacco-smoking signatures." It is not clear to me that there is a statistical test showing that the APOBEC-TP53 or APOBEC-PIK3CA positive association is significantly different depending on smoking status (or on the smoking signature SBS4/SBS92), and similarly so for the APOBEC-KRAS negative association. There would need to be a test for interaction between smoking status or intensity, which does not seem performed, to be able to claim "Indeed, we found that the interplay between the APOBEC and tobacco smoking mutagenesis has significant implications for tumor evolution".

Response:

We acknowledge the reviewer's concern regarding the statistical testing of interactions between APOBEC and tobacco smoking mutagenesis. While we recognize the potential limitations in statistical power due to the sample sizes (see the following table for details on the number of samples grouped by gene, APOBEC subtype, and smoking variables such as TTFC), we have now conducted formal statistical tests as suggested.

To evaluate whether the associations between APOBEC subtypes and driver mutations in *KRAS*, *TP53*, and *PIK3CA* depend on smoking variables, we used TTFC (Time to First Cigarette) as the primary smoking variable, as it was found to be the most relevant smoking metric associated with genomic features in our study. Specifically, we conducted logistic regression analyses using the following model in R:

```
glm(Driver Gene Mutation Status ~ APOBEC_Subtype * TTFC + Age + Sex + Histology + Tumor_Purity, family = "binomial")
```

Our results revealed a significant interaction between TTFC and APOBEC subtypes for *TP53* mutations ($P_{\text{interaction}} = 0.046$), indicating that the association between APOBEC mutagenesis and *TP53* mutations is influenced by smoking intensity. However, no significant interactions were observed for *PIK3CA* ($P_{\text{interaction}} = 0.26$) or *KRAS* ($P_{\text{interaction}} = 0.93$). We also tested interactions using alternative smoking variables (e.g., CIGT_PER_DAY) and observed consistent results (*TP53*: $P_{\text{interaction}} = 0.043$, *PIK3CA*: $P_{\text{interaction}} = 0.29$, *KRAS*: $P_{\text{interaction}} = 0.93$).

While we did not find significant evidence of interaction for *PIK3CA* and *KRAS* mutations, the significant interaction for *TP53* highlights the relevance of smoking mutagenesis in shaping APOBEC-associated driver mutations. These findings are now included in the revised manuscript, focusing on the interaction results for *TP53* mutations.

	TTFC Catalog	LAS			HAS		
		Total Sample	Mutated Sample	Mutation Frequency	Total Sample	Mutated Sample	Mutation Frequency
KRAS	>60mins	13	3	23.1%	18	4	22.2%
	31-60mins	26	6	23.1%	24	2	8.3%
	6-30mins	37	19	51.4%	24	10	41.7%
	<=5mins	36	9	25.0%	17	4	23.5%
PIK3CA	>60mins	13	1	7.7%	18	3	16.7%
	31-60mins	26	0	0.0%	24	3	12.5%
	6-30mins	37	2	5.4%	24	1	4.2%
	<=5mins	36	1	2.8%	17	0	0.0%
TP53	>60mins	13	6	46.2%	18	13	72.2%
	31-60mins	26	12	46.2%	24	20	83.3%
	6-30mins	37	19	51.4%	24	15	62.5%
	<=5mins	36	24	66.7%	17	11	64.7%

Finally, the presentation in the manuscript at various points can be improved:

- heavy speculation "Moreover, circadian disruption has previously been implicated in lung cancer tumorigenesis. Both catecholamine and glucocorticoid secretions are regulated by the circadian rhythm, and evidence suggests that their release is accelerated by components of cigarette smoke. When nicotine binds to nicotinic acetylcholine receptors expressed on adrenal medulla cells, it triggers the release of catecholamines into the bloodstream, which may promote epithelial cell proliferation, contributing to cell malignant transformation."

Response: We appreciate the reviewer's observation and acknowledge that the phrasing of these sentences could appear overly speculative, with insufficient elaboration on the supporting evidence. However, we would like to clarify that these sentences were not part of the submitted manuscript. They were included as a potential explanation in response to a previous reviewer's comment regarding the observed finding that "Tobacco smoking addiction (TTFC) is associated with genomic changes."

We recognize that circadian disruption may provide a plausible framework for interpreting our observations. Nevertheless, further experimental validation would be essential before incorporating this hypothesis into the manuscript.

- generic statements, devoid of specific or convincing information "every analysis we have conducted follows logical steps to verify our hypothesis and explain the results. We used multiple approaches and data types to prove our hypothesis. This study is based on correlation but the large number of consistent results across approaches substantiate our claim."

Response: We acknowledge that our previous response contained generic statements and lacked specific details. We intended to convey that the consistent findings across different types of omics data support the robustness of our conclusions. For instance, the associations between genomic (WGS data) and epigenomic (methylation data) features with TTFC were consistently observed. Additionally, potential cellular differences between HAS and LAS were identified using orthogonal methods (COO analysis using WGS data, pathway analysis and expression of cell markers based on RNA-Seq) reinforcing the validity of our observations. We recognize the importance of further experimental validation to strengthen our claims; however, we believe the results presented in our study, particularly as illustrated in Figure 5, provide strong evidence supporting our conclusions.

- unclear implications of TP53 mutations on their model

... "We do not emphasize the role of TP53 in reducing or activating apoptosis directly in our model, although several papers do show that p53 mutants retain partial function and can induce apoptosis under certain conditions (PMID: 16543937; PMID: 9841917; PMID: 11790556; PMID: 11058599)." TP53 mutants may retain partial activity, however certainly less so than the wild-type TP53 (or, they can have a dominant-negative function thus additionally poisoning also the wild-type allele).

Response:

We hypothesized that high levels of DNA damage—arising from unrepaired tobacco-induced lesions and elevated A3A-mediated mutagenesis—can trigger a senescence or an apoptotic response in HAS tumors. In the presence of *TP53* mutations (enriched in HAS tumors), the overall reduced p53 functionality can allow cells to escape senescence checkpoints and proceed to a crisis phase, potentially culminating in spontaneous mitotic arrest and cell death (PMID: 9841917; PMID: 11790556; PMID: 16543937; PMID: 26108857).

However, we acknowledge that the p53 function (with and without *TP53* mutations) is extremely complex since it controls a wide range of signaling networks. *TP53* mutants may have a dominant negative effect but also be converted to oncogenic proteins via gain-of-function (PMID: 29099487, PMID: 30538286, PMID: 32722796), further increasing DNA damage. Thus, also in line with the reviewer's concern for too much speculation, we opted to simplify our statement simply reporting how the increased DNA damage due to high mutagenesis and *TP53* mutant-induced genomic instability can trigger cellular senescence and apoptosis in HAS tumors.

..."Mechanistically, single-cell transcriptomics analyses in mice models have shown that during LUAD evolution, p53 promotes AT1 differentiation through action in a transitional cell state. Notably, p53 also directs alveolar regeneration after injury by regulating AT2 cell self-renewal and promoting transitional cell differentiation into AT1 cells. Thus, p53 governs alveolar differentiation (PMID: 37468633) and this can contribute to the higher stem-like phenotype of the HAS (TP53-enriched) tumors." If functional p53 is supposed to contribute to higher stem-like phenotype -- as based on their literature survey -- this does not seem to fit with that they observe HAS to have mutated (dysfunctional) TP53? HAS cannot be said to be TP53-enriched if they are TP53 mutant, in fact just the opposite is true. This may be an inconsistency in their logic, or simply a wording issue that could be resolved by rephrasing.

Response: We appreciate the reviewer's comment regarding the apparent inconsistency. Our intent is not to imply that mutant *TP53* directly imparts stem-like properties by performing the same functions as wild-type p53. Instead, our reasoning is that wild-type p53 normally guides alveolar cell differentiation and regeneration after injury. In doing so, it helps maintain tissue homeostasis and progression toward fully differentiated cell states. When *TP53* is mutated, these regulatory mechanisms are disrupted. Without the proper p53-driven differentiation cues, alveolar cells may fail to mature as intended, remaining in a more undifferentiated, "stem-like" state. Therefore, rather than suggesting that mutant *TP53* promotes stemness through the same mechanisms as wild-type p53, we are positing that the loss of p53's normal function can indirectly lead to an accumulation of less differentiated cells. We have rephrased these sentences in our manuscript to emphasize that it is the disruption of p53's usual differentiation program that contributes to the stem-like phenotype observed in the HAS (TP53-mutant) tumors.

- "Instead, we suggest that hypermutation by A3A and tobacco smoking mutagenesis, combined with limited DNA repair capacity and TP53-induced genomic instability, can trigger senescence, apoptosis, and cell regeneration" It was not clear to me where was the evidence for limited DNA repair capacity? Additionally, if anything, the A3A/A3B hypermutation has been linked to higher DNA repair activity by the HR, and also BER and MMR pathways in multiple studies.

Response:

We do not have evidence for limited DNA repair capacity. Our initial reasoning was based on the assumption that the hypermutation burden induced by both A3A activity and tobacco smoking might exceed the cell's repair capacity. We apologize for the lack of clarity in our previous response and appreciate the reviewer pointing out the studies

linking A3A/A3B hypermutation to higher activity in the HR, BER, and MMR pathways. We have revised our statement accordingly.

- "the most significant probe (AHRR cg05575921) we identified is the very well-known CpG site associated with smoking cessation (PMID: 28100713). Thus, we concluded that these findings reflect true biological differences, such as increased cell turnover" It does not seem immediately clear how a significant CpG probe known to be associated with smoking cessation implies there is increased cell turnover.

Response:

This comment pertains to our previous response to reviewers regarding the consistency of results from multiple data types in indicating how the combination of tobacco smoking and high APOBEC mutagenesis influences high cell turnover through cell death and cell regeneration. Being newly regenerated, many cells in HAS do not suffer from the effects of tobacco smoking; for example, they do not show methylation changes of the well known *AHRR* CpG probe, which are instead clearly present in LAS tumors. Thus, we are not claiming that the *AHRR* CpG probe is associated with increased cell turnover; rather, that the lack of smoking-related effects on *AHRR* in HAS tumors is the consequence of high cell death and related cell regeneration. In the manuscript, we state:

"These findings further support the hypothesis that the dynamic cell composition in HAS tumors can indirectly disrupt the reversion of methylation levels in cg05575921 (AHRR) following smoking cessation."

We apologize for any confusion caused by our earlier statement and appreciate the opportunity to clarify.

Reviewer #4 (Remarks to the Author):

Zhang et al provide a description of a large novel WGS dataset and an interesting analysis related to differences between lung tumors with low or high APOBEC activity. They have substantially revised the manuscript in general and addressed the concerns of the previous reviewers. Overall, this is an impactful study that warrants publication in Nature Communications. I only have a small number of additional points and questions.

Major points:

There are some discrepancies between SigProfiler/NMF and P-MACD results for the APOBEC activity which adds some confusion. It appears that the LAS/HAS calls are

derived from the SigProfiler results based on Figure 1A. Additional APOBEC mutations were identified in the LAS with the more sensitive P-MACD method which is specifically tailored towards characterization of APOBEC signatures. However, based on the P-MACD results in Supplementary Figure 3C,D, it is not clear that there are indeed two distinct groups of APOBEC samples. I suspect that the reason to keep the P-MACD results are to show that the LAS tumors were enriched in A3B-like mutations. But then it calls into question the overall grouping because some of the tumors in the LAS group have a higher number of APOBEC related mutations compared to some other tumors in the HAS group. While there is a significant shift in the mean, there is quite substantial overlap. If the LAS versus HAS grouping is made using the more sensitive P-MACD method, do the remaining results of the manuscript largely hold true?

Response: We appreciate the reviewer's insightful comment regarding the discrepancies between SigProfiler/NMF and P-MACD results for APOBEC mutagenesis and the potential implications for the LAS/HAS classification. Both SigProfiler/NMF and P-MACD are known methods for analyzing APOBEC mutational signatures, but they have distinct strengths and limitations. SigProfiler/NMF focuses on identifying overall mutational patterns in the cancer genome but has a detection threshold of approximately 5% for specific signatures, as noted in our previous studies on lung cancer (PMID: 34493867; Medrxiv: doi: <https://doi.org/10.1101/2024.05.15.24307318>). In contrast, P-MACD relies on motif enrichment analysis, making it more sensitive for identifying APOBEC signatures, but it does not provide precise estimates of APOBEC mutation load.

As the reviewer observed, our original analysis indicated that many tumors in the LAS group detected APOBEC mutations by P-MACD; however, the APOBEC mutation load estimated by P-MACD was generally lower than that assigned using SigProfiler. To address the reviewer's question, we reanalyzed the data using the more sensitive P-MACD method (see **Response Fig. 1**). Based on the distribution of roughly estimated minimal APOBEC mutation load, we used a threshold of $\log_2(\text{APOBEC_MutLoad_MinEstimate} + 1) = 10.5$ to redefine the LAS and HAS groups. This reclassification identified 182 LAS tumors and 127 HAS tumors. Notably, 84.5% of tumors were classified into the same APOBEC subtypes as in the original classification (88.5% for LAS and 79.3% for HAS).

Despite some reclassifications, the majority of our key findings remained consistent. For example:

- **Enrichment of Driver Gene Mutations:** *TP53* and *PIK3CA* mutations were significantly enriched in HAS tumors, while *KRAS* mutations were significantly enriched in LAS tumors.

- **APOBEC Mutagenesis:** HAS tumors were dominated by A3A-like mutagenesis, whereas LAS tumors were enriched in A3B-like mutagenesis.
- **UNG-APOBEC3B Expression Association:** Expression of *UNG* was significantly associated with *APOBEC3B* expression, but not *APOBEC3A*, in both HAS and LAS tumors.
- **Association between TMB/Methylation and Smoking Variables:** TTFC (time to first cigarette) was the only smoking-related variable significantly associated with tumor mutational burden, with this association observed exclusively in LAS tumors. Similarly, methylation levels of *AHRR* CpG sites tended to increase in former smokers compared to current smokers, with this trend apparent in LAS tumors but not in HAS tumors.
- **Cell Marker Expression:** LAS tumors showed significantly higher expression of AT2 cell markers, whereas HAS tumors exhibited higher basal cell marker expression.
- **Age at Diagnosis and Neoantigens:** HAS tumors were associated with significantly older age at diagnosis, particularly among heavy smokers (TTFC < 5 minutes). Within HAS tumors, APOBEC mutations generated more neoantigens than tobacco smoking, potentially contributing to increased tumor immune editing and a later age at diagnosis.

In summary, our results remain robust under the revised LAS/HAS classification using the P-MACD method, albeit with slightly less significant p-values in some cases. However, as mutational signature analysis becomes increasingly routine in cancer genomics, and given the established utility of SigProfiler or other NMF algorithms in previous studies, we believe it is more appropriate to retain our original method for classifying LAS and HAS tumors in this manuscript. Additionally, as pointed out by Reviewer #3, the use of SigProfiler/NMF for identifying APOBEC mutational signatures has been extensively reported and highlights its potential utility as a marker for immune therapy response.

The more “proximal-like” tumors from the COO analyses may refer to the Terminal Respiratory Unit (TRU) expression subtype of LUAD which were described in the TCGA marker paper (PMID: 25079552). This was formerly called bronchioid when first characterized (PMID: 17075127). This subtype tends to be enriched in EGFR rather than KRAS mutations.

Response:

In our cell-of-origin analysis, we aim to distinguish tumors originating from AT1/2 cells in the distal airway versus basal, club, or ciliated cells in the proximal airway, such as trachea, bronchi, and bronchioles. Recent studies have shown that epithelial cells can be

reprogrammed toward diverse lung cancer fates when exposed to the appropriate set of driver mutations. In the TCGA study, the terminal respiratory unit (TRU, formerly referred to as "bronchioid") was defined based on transcriptional and epigenetic subtypes, correlated with histopathological, anatomical, and mutational classifications. However, these classifications may not reflect specific cell types within the anatomical compartments of the airway.

In our study, we observed that tumors with proximal airway cell characteristics (*e.g.*, with high expression of basal cell markers) are associated with high APOBEC signature (HAS) tumors, which are not enriched in *KRAS* mutations. Moreover, the frequency of *EGFR* mutations in our cohort is relatively low (4.5% overall, with no statistically significant differences between HAS (3.0%) and LAS (5.8%) tumors (Fisher's exact test, $P = 0.5$) or between tumors classified as proximal-like versus distal-like based on cell type markers ($P = 1$). Thus, our classification is different from the TRU expression subtype in the original TCGA LUAD study.

Minor points:

I found this line in the abstract to be confusing because the inclusion of both previous literature and current findings: "Hypermethylation by unrepaired A3A and tobacco smoking mutagenesis combined with TP53-induced genomic instability can trigger senescence⁷, apoptosis⁸, and cell regeneration⁹, as indicated by high expression of pulmonary healing signaling pathway, stemness markers and distal cell-of origin in HAS." Maybe a clearer way to state this is something like "Previous studies have shown that hypermethylation by unrepaired A3A and tobacco smoking mutagenesis combined with TP53-induced genomic instability can trigger senescence⁷, apoptosis⁸, and cell regeneration⁹. We observed that HAS exhibited high expression of pulmonary healing signaling pathway, stemness markers and distal cell-of origin."

Response: We appreciate the insightful feedback. We have revised the text accordingly.

Also in the abstract, the phrase "newly generated, unmutated cells" is not standard and not used in the manuscript. It is confusing because "unmutated cells" sounds like it is referring to normal cells but the results and conclusions are discussing cancer cells. Can you rephrase?

Response: Again, we thank the reviewer for pointing out this possible confusion. We have revised the wording as “**newly generated progenitor-like cells**” which we believe more accurately reflects the context of our findings and aligns with the results and conclusions discussed in the manuscript.

Point-by-point responses to Reviewer #5's comments relative to questions from Reviewer #3

Reviewer #4 (Remarks to the Author):

The authors have addressed by concerns and questions

Reviewer #5 (Remarks to the Author):

Regarding comments by Reviewer #3:

In his first four comments Reviewer #3 asks to place findings from the current study in light of existing literature ("An important issue for authors to seriously consider is presenting various findings as novel either explicitly or having context suggesting that, while they were reported before."). Authors have now added the references suggested by the reviewer, thus responding to the specific examples given. However there seem to be additional instances, where more nuanced language could be helpful to provide findings of the current study in the light of existing literature. Increased tumor heterogeneity has e.g. also been reported in connection with APOBEC in lung cancer (Roper et al., Cell Rep 2019). Overall, many findings presented are note entirely novel and the field would benefit from citing relevant sources and use language that acknowledges when findings validate existing data even though this may limit novelty.

Response:

We appreciate the reviewer for the additional comments regarding the novelty of our findings in the context of the existing literature. We acknowledge the importance of the reference reported (Roper et al., Cell Rep 2019), which showed that APOBEC mutagenesis—driven by increased expression of the APOBEC3 region transcripts and associated with a high-risk APOBEC3 germline variant—correlates with mutational tumor heterogeneity and contributes to proteogenomic tumor evolution in metastatic thoracic tumors. We have now cited this key study in both the *Introduction* and *Discussion* sections of our manuscript to properly acknowledge previous findings while highlighting the unique contributions of our study (see below).

In the Introduction, we added:

"These signatures have been associated with mutational tumor heterogeneity (Roper et al., 2019) and an improved response to immunotherapy."

In the Discussion, we added:

"Previous studies suggest that APOBEC mutagenesis contributes to proteogenomic tumor evolution and heterogeneity in metastatic thoracic tumors (Roper et al., 2019). Here, we further refine our understanding of how heterogeneity in mutational burden—driven by co-occurring mutational processes and diverse cell types—shapes clonal evolution, offering important insights into lung carcinogenesis."

Additionally, we reviewed recently published literature from the past few months and incorporated relevant new findings from an additional pre-print manuscript from our group based on animal models.

In the Results section, we added:

"... This finding highlights the potential interplay between APOBEC activity and tobacco smoke-induced mutagenesis, an interaction also supported by a recent study on oral tumorigenesis in animal models (Durfee et al., 2025)."

We appreciate the reviewer's insights, which have helped us improve the contextualization of our findings within the existing literature.

In his next four comments, Reviewer #3 is concerned about the statistical rigor of the analyses, which appears to be broader despite giving a few examples: "there remain issues to be sorted out regarding statistical support of findings, which presents, essentially, anecdotal examples in place of rigorous, systematic tests. This is of course convenient for getting passable p-values/FDRs however they should consider a more rigorous approach. Examples thereof would be as follows..."

To the specific examples the authors respond adequately, even though a test for statistical interaction between APOBEC_subtype and TTFC done in response to comment (iii) is only significant for TP53, but not other genes. His comment, that without a statistical interaction test authors would not "be able to claim 'Indeed, we found that the interplay between the APOBEC and tobacco smoking mutagenesis has significant implications for tumor evolution'." does not seem to be corroborated conclusively and Reviewer #3 would potentially reject this statement in its current form.

Response:

We understand this reviewer remains concerned about the claim regarding the interplay between APOBEC and tobacco smoking mutagenesis. In our previous response, we provided results from an association analysis between APOBEC subtypes and driver mutations in *KRAS*, *TP53*, and *PIK3CA*, using TTFC (Time to First Cigarette) as the primary smoking variable to test the interaction with APOBEC subtypes. Our results

revealed a significant interaction between TTFC and APOBEC subtypes for *TP53* mutations ($P_{\text{interaction}} = 0.046$); however, no significant interactions were observed for *PIK3CA* or *KRAS*.

As noted in our previous response, these analyses may have potential limitations in statistical power due to sample size constraints. Overall, *TP53* mutations were the most frequent in our dataset (61%), while *KRAS* (29%) and *PIK3CA* (6%) mutations were significantly less common. The lower mutation frequencies for *KRAS* and *PIK3CA* may have reduced our power to detect statistically significant interactions between APOBEC activity and TTFC.

To address the reviewer's concerns, we have now explicitly stated this limitation in our *Results* and *Methods* sections and have adjusted the wording of our conclusions accordingly. Additionally, we have incorporated findings from a recent study (PMID: 39896515), which provides further support for the interplay between APOBEC and tobacco smoke-induced mutagenesis. This study employed a mouse model to investigate oral tumorigenesis driven by mutations from both the endogenous mutagen (human A3B) and the exogenous tobacco surrogate (NQO, 4-nitroquinoline 1-oxide). These results reveal a synergistic interaction between these two mutational processes, suggesting that tobacco smoke carcinogens can exacerbate APOBEC mutagenesis and DNA damage. While this study did not focus specifically on driver mutations, it provides strong evidence for the broader interplay between APOBEC activity and tobacco smoke-induced mutagenesis in cancer. We have now cited this study in our manuscript.

Revision in our manuscript:

"In addition, we observed a significant interaction between TTFC and APOBEC subtypes ($P_{\text{interaction}} = 0.046$) when assessing whether TTFC affects the relationship between APOBEC mutagenesis and driver gene mutations in TP53. This finding highlights the potential interplay between APOBEC activity and tobacco smoke-induced mutagenesis, an interaction also supported by recent studies on oral tumorigenesis in animal models (Durfee et al. 2025)."

However, concerns by reviewer #3 appear to be more general about minimizing the risk of overinterpreting results of statistical tests that may not be well controlled. While examples given by reviewer #3 have been addressed, several more exist, for example:

- In ext. Figure 7b a borderline significant difference was observed for CD4 Th1 cells in LAS vs. HAS only, after doing nine two-group comparisons. Is this p adjusted for multiple testing? Ext. Fig. 7c shows no difference in lymphocytes. Claiming differences in immune infiltration in the text would require validation eg by histology.

Response:

For Extended Data Fig. 7b, the p-value was not adjusted for multiple testing. We agree that claiming significant differences in immune infiltration in the text would require additional validation. To address the reviewer's concerns and minimize the risk of overinterpretation, we have removed Extended Data Fig. 7b and 7c, along with the corresponding results and methods, from our manuscript.

- It is not clear whether/how the regression tests between mutational load and cigarette smoking were adjusted for multiple testing in Fig. 3b, Suppl. Fig 10 and Ext. Data Fig. 5. Correlating 5 cigarette-related variables against mutation load in two groups yields ten comparisons (unless each level per variable was tested individually). Was the family-wise error rate controlled?

We thank the reviewer for the insightful comment. The figures referenced (Fig. 3b, Supplementary Fig. 10, and Extended Data Fig. 5) present similar multivariate regression analyses assessing the relationship between smoking variables and mutational burdens. Specifically, Fig. 3b examines total mutational burden (TMB), Extended Data Fig. 5 includes SBS4, DBS2, and ID3, while Supplementary Fig. 10 focuses on TMB but is limited to LUAD samples.

Using Fig. 3b as an example, as described in the Methods, we performed multivariate regression analyses to evaluate the independent effects of five categorical smoking variables on genomic mutation burden (TMB), adjusting for additional potential confounders (age, sex, histology, and tumor purity). These five smoking-related variables were included in the same model because they are conceptually related but not strongly correlated with each other. The regression model used is as follows (in R programming):

```
lm(TMB ~ CIGT_TOT_DURATION + TTFC + CIGT_PER_DAY +  
CIGT_AGE_1BEGIN + CIGT_TIME_LAST_QUIT + Age + Sex + Histology +  
Tumor_Purity, data=.)
```

We repeated this analysis separately for LAS and HAS tumor groups to assess subgroup differences. Fig. 3b presents a forest plot summarizing the effect sizes and p-values for each smoking variable, allowing for a visual comparison between LAS and HAS. Since all five smoking variables were included in the same regression model, multiple testing correction is not necessary within each model, as the p-values for these variables are already adjusted for the inclusion of other smoking-related variables. However, we acknowledge that we originally did not control for multiple comparisons across the two groups (LAS vs. HAS) for the p-values within the forest plots, since each level per variable was tested individually as the reviewer noticed.

To further strengthen our analysis, we also performed trend testing, treating the categorical smoking variables as continuous variables. We specifically highlighted the P-trend values, and TTFC was the only significant smoking variable associated with TMB. In response to the reviewer's concern about family-wise error rate (FWER) control across the LAS and HAS comparisons, we have now added FDR-adjusted p-values for TTFC in all relevant figures. The conclusions remain unchanged. For example, all significant associations were observed only in LAS tumors, not HAS tumors. Also, TTFC was the only smoking variable significantly associated with TMB, even after controlling for FWER.

To ensure clarity, we have also updated Supplementary Fig. 10 and Extended Data Fig. 5 to include FDR values for all TTFC associations and have revised the Methods section to clearly describe these analyses.

- Were cell-type markers in 4a adjusted for multiple testing and how were these markers selected?

Response:

The LUAD cell type-specific markers ($N = 52$) referenced in our study were initially derived from a recent single-cell RNA-Seq analysis of LUAD (PMID: 34764257). Based on feedback from the previous review, we expanded our analysis to include a more comprehensive set of cell markers for human AT2 and basal cells, incorporating markers from the CellMark2.0 database (PMID: 36300619; $N = 113$). As shown in Supplementary Fig. 16c, these markers exhibit the same trend in association with APOBEC subtypes.

Figure 4a presents a representative selection of well-established AT2 and basal cell markers to illustrate APOBEC subtype associations. In our original submission, multiple testing adjustments were not applied to the p-values in Fig. 4a. In response to the reviewer's comment, we have now included FDR-adjusted p-values to account for multiple testing across the collected cell markers.

- For the pathway enrichment (Suppl. Fig. 16), gene expressions were correlated to TTFC within the HAS or LAS group independently. This is surprising, considering that smoking is reported as leaving little mutational and epigenetic impact in HAS tumors and that no correlation of TTFC with mutation load etc. is observed in HAS tumors. "Pulmonary healing" shows strongest association with Time To First Cigarette in HAS tumors, which fits the proposed mechanisms of quiescence and stemness. This pathway is however also upregulated in LAS tumors, albeit to a lower degree, while many other pathways are upregulated in HAS, but suppressed in LAS tumors (e.g.

PDGF signaling, Role of pattern recognition pathways). These may actually hold more insights into biological differences between LAS and HAS tumors than “Pulmonary healing”. A more rigorous and unbiased approach to identify processes that are different between HAS and LAS tumors would be gene set/pathway enrichment between LAS and HAS expression rather than correlation of TTFC within each group, especially as no relevant dose-response relationships of TTFC with other parameters are observed in HAS tumors.

Response:

We thank the reviewer for the insightful comment. We agree that a more rigorous and unbiased approach is essential to identify processes that distinguish LAS and HAS tumors. As noted in our previous response to Reviewer #3, we have already performed this analysis and highlighted the consistent findings. Specifically, we conducted differentially expressed gene (DEG) analysis followed by IPA pathway analysis on the most significantly differentially expressed genes between LAS and HAS tumors.

The IPA results identified key pathways associated with molecular and cellular functions, including "Cell Death and Survival," "Cellular Function and Maintenance," and "Cellular Development." These findings align with the expected upregulation of these processes in HAS tumors, as described in our manuscript. For example, among the top differentially expressed genes:

- PDCD10 (Programmed Cell Death 10, $P = 1.73e-05$): Directly involved in programmed cell death and survival signaling.
- PRDX1 (Peroxiredoxin 1, $P = 5.4e-06$): Protects cells from oxidative stress-induced apoptosis.
- IL18 (Interleukin 18, $P = 1.84e-05$): Modulates inflammatory responses and cell death pathways.

To further refine our analysis, we conducted Gene Set Enrichment Analysis (GSEA) using Hallmark gene sets (MSigDB, category “H”) to compare transcriptomic profiles between HAS and LAS tumors. Genes were ranked by log₂ fold-change from the differential expression analysis (HAS vs. LAS), and we identified multiple significantly enriched gene sets (q -value < 0.05).

Key enriched pathways in HAS tumors included:

- Cell cycle progression and proliferative signaling: HALLMARK_E2F_TARGETS, HALLMARK_MYC_TARGETS_V1, and HALLMARK_G2M_CHECKPOINT, suggesting increased proliferative capacity and potential for cell regeneration.

- Growth and developmental pathways: HALLMARK_MTORC1_SIGNALING and HALLMARK_WNT_BETA_CATENIN_SIGNALING, pointing to altered regulation of tumor growth and differentiation.
- Immune and inflammatory signaling: HALLMARK_INFLAMMATORY_RESPONSE, HALLMARK_INTERFERON_GAMMA_RESPONSE, and HALLMARK_ALLOGRAFT_REJECTION, indicating a more active immune microenvironment in HAS tumors.
- Metabolic and stress response pathways: HALLMARK_GLYCOLYSIS, HALLMARK_OXIDATIVE_PHOSPHORYLATION, HALLMARK_UNFOLDED_PROTEIN_RESPONSE, and HALLMARK_APOPTOSIS, reflecting metabolic reprogramming and adaptive stress responses.

These consistent findings between IPA and GSEA suggest that HAS tumors exhibit enhanced proliferative signaling, immune/inflammatory activity, and metabolic adaptation, providing critical insights into the biological underpinnings of HAS tumorigenesis. We have incorporated the GSEA results in the manuscript (now Supplementary Figure 16d).

In the remaining comments, Reviewer #3 seems to be concerned with the description and presentation of results, which in some parts lack support by the study's data. He also provides several examples of "heavy speculation" or proposition of potential biological mechanisms and claims that would require additional evidence to be valid conclusions, e.g. regarding the role of TP53, the extent of DNA damage or mechanism of AHHR methylation. The authors respond to all points and have reduced the amount of speculation around these issues.

Response:

Thank you for recognizing our efforts in reducing the amount of speculation identified in the previous reviews.

In general, the manuscript may benefit from use of more neutral language and reduced claims regarding causal effects and overinterpretation of the results regarding molecular processes and mechanisms. Some examples from the title and abstract only are:

o 'APOBEC shapes tumor evolution and age at onset of lung cancer' implies causality, but this retrospective study only shows correlations between high APOBEC mutation

load and tumor onset age. Age might just as easily influence APOBEC activity, or a third factor could protect some individuals by clearing smoking-related mutations. In that scenario, only those with high APOBEC ever develop tumors (and then later), while those with low APOBEC never do. Thus, no causal directions can be established here. The study illuminates these associations but cannot prove causation; experimental or Mendelian randomization evidence would be needed to confirm any causal links.

Response:

The reviewer is correct in arguing that we did not conduct formal experiments to prove causality. However, as we extensively discussed in the manuscript, we have multiple orthogonal confirmations of our hypothesis of the effect of the interplay of tobacco smoking with APOBEC, as evidenced also in melanoma through UV exposure and in mouse oral cancer through smoking carcinogens. In response to the reviewer's concern, we toned down any statement that was not experimentally proven throughout the manuscript, moving data interpretations to the Discussion section. Moreover, we have substituted "shapes" with "affects", which is a non-deterministic term, even in the title.

o In "The expected association of tobacco smoking variables (e.g., time to first cigarette) with genomic/epigenomic changes are not observed in HAS, a plausible consequence of frequent cell senescence or apoptosis.", the second part is speculative as there is no evidence showing actual differences in senescence or apoptosis in the High APOBEC Signature (HAS) group in the study. It may thus be placed in the discussion rather than the abstract.

Response:

We agree with the reviewer's suggestion and have removed "a plausible consequence of frequent cell senescence or apoptosis" from the abstract, retaining it only in the discussion."

o Also, authors state that there may be "...frequent immuno-editing" in the HAS group. Typically, immune editing is defined as the selection of neoantigen-depleted clones (e.g. Zapata et al., Nature Gen 2023; Roerden & Spranger, Nat Rev Imm 2025), which leads to lower neoantigen load and ITH compared to tumors with immune escape/evasion. However, they observe more neoantigens in HAS (but no differences in immune infiltration if controlling for multiple testing), which seems to be the opposite direction as implied by the statement in the abstract.

Response:

We agree with the reviewer that our statement may be misleading, especially given the lack of significant immune infiltration results after multiple testing. Therefore, we have removed 'frequent immuno-editing' from the abstract as well as its interpretation from the results section.

Similarly, clarity of the manuscript may benefit from a more stringent distinction of describing current results and interpretation by moving statements into the discussion that are not derived from study data. Examples of statements/paragraphs in the results section that largely depend on previously known results and may be better located in the discussion section are:

o “The frequent episodic APOBEC mutagenesis could explain the lack of association between APOBEC3A and UNG/BER expression. However, if tumors experienced frequent APOBEC bursts, we would see APOBEC mutations at different times, including during subclone expansion. The fact that many samples, particularly HAS tumors, exhibit only clonal mutations, does not support this hypothesis. Moreover, it is unlikely that chronologically discrepant episodic APOBEC3A mutagenesis and UNG/BER induction show similar findings across all cancer types and multiple BER enzymes.”

Response:

We thank the reviewer for this suggestion. We agree that it is more appropriate to include these sentences in the Discussion. Therefore, we have moved them to the Discussion section where we interpreted the lack of association between *APOBEC3A* expression and the mRNA expression of *UNG* and other *BER* genes in our study.

o “A more plausible reason for this finding is the different cell composition between HAS and LAS tumors. Strong DNA damage induced by APOBEC3A hypermutation and TP53-associated genomic instability can cause more cell senescence and apoptosis^{52–55} in HAS tumors. Apoptosis and senescence can in turn lead to cell regeneration^{56,57}. HAS tumor composition likely includes a high number of newly generated or de-differentiated cells, which do not display the expected tobacco smoking associated patterns that are evident in the more differentiated LAS tumors.” Additional histological and/or experimental evidence would be needed to conclude this from the study.

Response:

We agree with the reviewer's comments. Without direct experimental evidence, it is more appropriate to present this interpretation in the discussion. Therefore, we have moved these sentences from the results to the discussion section, where we explore how the dynamic cellular state and composition in HAS could explain the lack of

expected associations between tobacco smoking exposure and genomic or epigenomic changes in this subtype.

o “The neoantigens in HAS tumors likely increased tumor immune-editing during tumor development⁷², potentially contributing to their late age at diagnosis.” Differences in immune activity are not shown between LAS and HAS tumors.

Response:

As mentioned earlier, we have removed this sentence related to immune-editing from our manuscript.